# MatRIS: Toward Reliable and Efficient Pre-trained Machine Learning Interatomic Potentials

**Yuanchang Zhou**[1,2]    **Siyu Hu**[1,2]    **Xiangyu Zhang**[1,2]    **Hongyu Wang**[1,2,3]

**Guangming Tan**[1,2] [*]    **Weile Jia**[1,2,*]

[1]State Key Lab of Processors, Institute of Computing Technology, Chinese Academy of Sciences

[2]University of Chinese Academy of Sciences

[3]School of Advanced Interdisciplinary Sciences, UCAS

{zhouyuanchang23s, jiaweile}@ict.ac.cn

## Abstract

Foundation MLIPs demonstrate broad applicability across diverse material systems and have emerged as a powerful and transformative paradigm in chemical and computational materials science. Equivariant MLIPs achieve state-of-the-art accuracy in a wide range of benchmarks by incorporating equivariant inductive bias. However, the reliance on tensor products and high-degree representations makes them computationally costly. This raises a fundamental question: as quantum mechanical-based datasets continue to expand, can we develop a more compact model to thoroughly exploit high-dimensional atomic interactions? In this work, we present MatRIS (**Mat**erials **R**epresentation and **I**nteraction **S**imulation), an invariant MLIP that introduces attention-based modeling of three-body interactions. MatRIS leverages a novel separable attention mechanism with linear complexity $O(N)$, enabling both scalability and expressiveness. MatRIS delivers accuracy comparable to that of leading equivariant models on a wide range of popular benchmarks (Matbench-Discovery, MatPES, MDR phonon, Molecular dataset, etc). Taking Matbench-Discovery as an example, MatRIS achieves an F1 score of up to 0.847 and attains comparable accuracy at a lower training cost. The work indicates that our carefully designed invariant models can match or exceed the accuracy of equivariant models at a fraction of the cost, shedding light on the development of accurate and efficient MLIPs.

## 1 Introduction

Quantum Mechanics (QM)-based calculations are the cornerstone of modern drug and material research, providing highly accurate modeling of interatomic interactions. However, their prohibitive computational cost makes large-scale simulations intractable (De Vivo et al., 2016; Jain et al., 2013b; Merchant et al., 2023). Machine learning interatomic potentials (MLIPs) have emerged as a powerful alternative, enabling accelerated, long-timescale molecular dynamics (MD) simulations while retaining near-quantum-chemical accuracy. With the increase in QM-based reference data and model innovations, MLIPs have demonstrated remarkable accuracy and generalization in property prediction and materials discovery (Merchant et al., 2023; Barroso-Luque et al., 2024; Zhang et al., 2024; Yang et al., 2024; Zhang et al., 2025; Fu et al., 2025; Wood et al., 2025).

Graph neural networks (GNNs) have been widely adopted for 3D molecular modeling, where atoms are represented as nodes and interatomic interactions as edges (Qu & Krishnapriyan, 2024; Liao & Smidt, 2023; Liao et al., 2024b; Fu et al., 2025). Through Message Passing (MP), node features are iteratively updated to capture local and global structural interactions. To enhance model expressiveness and generalization, many MLIPs incorporate domain-specific inductive biases (e.g., translation,

---

[*]Corresponding author

rotation, permutation, reflection invariance, or equivariance). Depending on how these symmetries are encoded, MLIPs are broadly divided into invariant, equivariant and unconstrained architectures (Duval et al., 2024; Jacobs et al., 2025). In invariant models, the structural descriptor is encoded based on attributes such as interatomic distances, bond angles, and dihedral angles (Gasteiger et al., 2020; 2022b; Klicpera et al., 2021; Deng et al., 2023; Zhang et al., 2025). In equivariant models, higher-order equivariance is typically enforced through computationally intensive tensor products of rotation order $L$ (Batzner et al., 2022; Batatia et al., 2022; Liao & Smidt, 2023; Liao et al., 2024b).

Previous work demonstrates that equivariant models often deliver superior accuracy (Batzner et al., 2022). In contrast, unconstrained models do not explicitly encode symmetries; instead, the model learns them from data or approximates them through data augmentation or auxiliary losses, leading to more flexible architectural design (Duval et al., 2024; Neumann et al., 2024; Rhodes et al., 2025). Meanwhile, the results in a popular benchmark (Matbench-Discovery leaderboard (Riebesell et al., 2025)) indicate that equivariant GNNs achieve higher accuracy. When the training data is MPTrj (Deng et al., 2023), eSEN-30M-MP (Fu et al., 2025) and eqV2 S DeNS (Barroso-Luque et al., 2024) reach F1 scores of 0.831 and 0.815, with energy errors of 0.033 and 0.036 meV/atom, respectively. Despite the accuracy gains, the heavy equivariant operations make equivariant methods significantly more computationally expensive and memory-intensive. Our investigation into the training cost of several

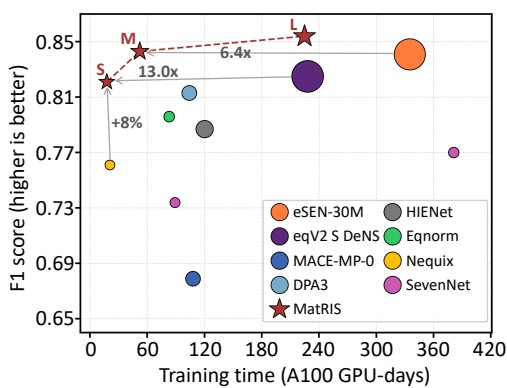

Figure 1: Trade-offs between training time and F1 score of foundation MLIPs. Training times for eSEN and eqV2 are estimated on Nvidia-A100. Nequix (Koker & Smidt, 2025) was trained on `JAX`; all others on `PyTorch`. Larger marks indicate models with more parameters.

mainstream pretrained models is summarized in Figure 1. eSEN-30M-MP and eqV2 S DeNS show superior F1 score while requiring 335 and 228 GPU days, respectively. This high computational demand can be caused by three main factors: 1) the intensive equivariant operations such as tensor products, 2) the large number of model parameters, and 3) the prolonged training schedules (e.g., 100, 150, and 600 training epochs are reported for eSEN-30M-MP, eqV2 S DeNS, and SevenNet-l3i5 (Kim et al., 2025), respectively).

Incorporating equivariance into the GNN MLIPs serves as an implicit form of data augmentation. AlphaFold has shown that, with sufficient data, non-equivariant models can accurately predict protein secondary structures (Jumper et al., 2021; Abramson et al., 2024). In MLIPs, the rapid increase in QM-based reference datasets (Barroso-Luque et al., 2024; Levine et al., 2025; Gharakhanyan et al., 2025; Sriram et al., 2025) motivates us to ask: **Is the equivariance indispensable as the QM-based dataset continues to increase? Can we develop a more compact architecture to capture the high-dimensional atomic interactions encoded in QM-based data sufficiently?**

We have the following findings: 1) Recent studies (Yang et al., 2024; Zhang et al., 2024; 2025) show that invariant models offer reliable property predictions and enable a wide range of scientific applications while maintaining computational efficiency. 2) For a more compact architecture (i.e., one that can fully exploit QM-based data). Element types and pairwise interactions have been shown to be insufficient for distinguishing graphs with different chemical properties (Xu et al., 2019). Incorporating three-body interactions is needed to exploit the knowledge in QM-based data. Self-attention mechanisms (Mazitov et al., 2025; Qu & Krishnapriyan, 2024) have proven to be a promising method in improving model expressiveness, also benefiting in model scalability.

Building on these insights, we introduce an invariant MLIP: interatomic potential for **Mat**erials **R**epresentation and **I**nteraction **S**imulation (**MatRIS**). To the best of our knowledge, our model is the first to explicitly leverage an $O(N)$ attention mechanism to model three-body interactions. MatRIS consists of graph generation, feature embedding, graph attention, refinement, and a readout block. We provide the ablation study of these modules in this paper. Putting all these modules together, MatRIS achieves state-of-the-art (SOTA) accuracy and efficiency across a wide range of

chemical applications. Additionally, the novel separable attention has lower complexity ($O(N)$) compared to full attention ($O(N^2)$). Across diverse benchmarks, MatRIS achieves competitive results. MatRIS-L achieves SOTA results on compliant Matbench-Discovery with an F1 of **0.847** and a root mean square displacement (RMSD) of **0.0717**. Moreover, MatRIS-S and MatRIS-M deliver accuracy comparable to eqV2 S DeNS and eSEN-30M-MP, respectively, while improving training efficiency by **13.0×** and **6.4×**, respectively. These results demonstrate MatRIS's strong potential for applications in materials science and drug discovery.

## 2 RELATED WORKS

**Invariant MLIPs.** Invariant MLIPs are models whose intermediate representations are invariant under rotations and translations (Duval et al., 2024). This invariance is achieved by using internal coordinates instead of Cartesian coordinates, with features such as interatomic distances, bond angles, and dihedral angles remaining unchanged under rotations and translations of the system (Schütt et al., 2017; Gasteiger et al., 2020; Novikov et al., 2020; Fan et al., 2022; Gasteiger et al., 2022b; Chen & Ong, 2022; Deng et al., 2023; Zhang et al., 2025). Early invariant MLIPs, such as SchNet (Schütt et al., 2017), CGCNN (Xie & Grossman, 2018), and PhysNet (Unke & Meuwly, 2019), employ relative distances between node pairs and encode local geometric information via learnable radial basis functions. More recent MLIPs enhance representational expressiveness by incorporating higher-order many-body scalar features. For example, the DimeNet (Gasteiger et al., 2020; 2022a) series introduced directional message passing, allowing angular information to be embedded in edge updates between atoms. The GemNet series (Gasteiger et al., 2022b; Klicpera et al., 2021) further incorporates dihedral angles to improve performance. SphereNet (Liu et al., 2022) and ComENet (Wang et al., 2022) proposed methods to efficiently extract four-body angles within local neighborhoods, avoiding the need to iterate over all three-hop neighbors. DPA3 (Zhang et al., 2025) builds upon the line graph series (LiGS), capturing higher-order interactions. Invariant MLIPs are progressively increasing their representational expressiveness while maintaining inherent computational efficiency. Building upon these insights, we design MatRIS from an invariant perspective. We further provide a detailed discussion in Appendix A, highlighting how MatRIS differs from other MLIPs that incorporate three-body encodings or attention-based mechanisms.

**Equivariant MLIPs.** Equivariant MLIPs are models where intermediate representations are invariant (e.g., scalars) or equivariant (e.g., vectors or higher-order tensors) under rotations (Duval et al., 2024). Current equivariant MLIPs can be divided into scalarization-based models (Schütt et al., 2021; Du et al., 2023; Thölke & Fabritiis, 2022; Aykent & Xia, 2025) and high-degree steerable models (Musaelian et al., 2023; Batzner et al., 2022; Batatia et al., 2022; Liao & Smidt, 2023). Scalarization-based MLIPs model interatomic interactions in the Cartesian coordinate system while restricting the set of operations on geometric features to preserve equivariance (Duval et al., 2024; Wang et al., 2024; Yin et al., 2025). On the other hand, high-degree steerable equivariant MLIPs use irreducible representations (irreps) to encode features, ensuring equivariance under 3D rotations. Each irrep of degree $L$ corresponds to a $(2L + 1)$-dimensional vector space (Batzner et al., 2022; Batatia et al., 2022; Liao & Smidt, 2023). In equivariant GNN-based MLIPs, MP involves transforming and combining these type-$L$ vectors. To interact across degrees during MP, tensor products (by using Clebsch–Gordan coefficients to combine) are employed. To avoid excessive computational complexity, these models typically employ only low-degree equivariant representations (Park et al., 2024; Liao et al., 2024b; Fu et al., 2025). Equivariant MLIPs continue to deliver SOTA accuracy on various benchmarks (Tran et al., 2023; Riebesell et al., 2025), while remaining computationally demanding.

**Unconstrained MLIPs.** Unconstrained MLIPs do not impose strict constraints on their intermediate representations. Instead, these models typically learn symmetries directly from the data or incorporate additional loss terms to encourage symmetry learning (Duval et al., 2024; Rhodes et al., 2025). For example, Qu & Krishnapriyan (2024); Neumann et al. (2024); Rhodes et al. (2025) use data augmentation (applying random rotations to training samples) to learn rotational equivariance and have demonstrated promising results. In addition, Neumann et al. (2024) enhances stability in MD simulations by removing net force and torque, while Rhodes et al. (2025) introduces an 'Equigrad' loss to incentivize rotational invariance of energy. Unconstrained MLIPs have inference efficiency comparable to invariant MLIPs and more flexible architectures. These models

demonstrate competitive accuracy in multiple benchmarks (Chanussot et al., 2021; Tran et al., 2023; Riebesell et al., 2025). However, studies indicate that they may lead to errors in certain property prediction tasks (Fu et al., 2023; Póta et al., 2025; Bigi et al., 2025).

## 3 MATRIS

In this section, we introduce the detailed architecture of MatRIS. The interaction between the atom graph and the line graph is described in Section 3.1. The Graph Attention module is depicted in Section 3.2. In Section 3.3, we describe other key components of MatRIS. An overview of MatRIS is shown in Figure 3, and the model formalizations are detailed in Appendix B.

### 3.1 LINE–ATOM GRAPH INTERACTION

To model three-body interactions, we explicitly construct a Line Graph. In the Atom Graph, nodes represent atom types and edges represent pairwise interactions (bonds), whereas in the Line Graph, nodes represent edges of the atom graph and edges represent three-body interactions (angles) (Harary & Norman, 1960; Whitney, 1992).

Specifically, given an Atom Graph $G^a = (V^a, E^a)$, where $V^a$ is the set of atoms and $E^a$ is the set of edges within a cutoff distance $r^a_{\text{cut}}$, the corresponding Line Graph $G^l = (V^l, E^l)$ is constructed as follows: ① Each node in $V^l$ corresponds to an edge in $E^a$; ② An edge $e \in E^l$ is added between two nodes if their corresponding edges in $E^a$ share a common atom, representing the angular information formed by the three atoms (i.e., the three-body interaction). The conversion from the atom graph to the line graph is illustrated in Figure 2.

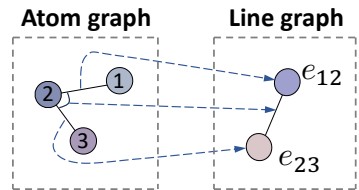

Figure 2: Conversion from an Atom Graph to a Line Graph.

For graph information fusion, we update the Line Graph to obtain edge and angular features that encode three-body interactions. These updated edge features are then propagated back to the Atom Graph, enabling the atomic features to incorporate higher-order information from the Line Graph.

### 3.2 GRAPH ATTENTION

In this section, we introduce the design motivation and implementation of the Dim-wise Softmax and Separable Attention mechanisms.

**Dim-wise Softmax.** Recent studies have shown that attention mechanisms play an important role in improving both the accuracy and scalability of MLIPs (Liao & Smidt, 2023; Liao et al., 2024b; Qu & Krishnapriyan, 2024). Existing approaches (Liao & Smidt, 2023; Wang et al., 2024; Shao et al., 2025; Liao et al., 2024b) typically compute attention weights $a_{id}$ to weight the value vectors $V \in \mathbb{R}^{\text{neighbors} \times D}$, where $D$ denotes the hidden dimension. The weights $a_{id}$ depend on the features of node $i$ and its neighbors $\mathcal{N}(i)$, while the values $V$ are obtained by applying a nonlinear transformation to the fused edge and node features.

In these methods, the same attention weights are applied to all feature dimensions, implicitly assuming equal importance across dimensions. However, this assumption limits the model's ability to distinguish the independent contributions of different feature dimensions. Our proposed Dim-wise Softmax computes attention scores independently for each feature dimension. Given an input feature $x \in \mathbb{R}^{\text{neighbors} \times D}$ and a neighbor list $\mathcal{N}$, the Dim-wise Softmax is computed as follows:

$$\alpha_{id} = \text{Dim-wise Softmax}(x_{id}, \mathcal{N}(i)) = \frac{\exp(x_{id})}{\sum_{k \in \mathcal{N}(i)} \exp(x_{kd})} \quad (1)$$

where $\alpha \in \mathbb{R}^{\text{neighbors} \times D}$ is the attention weight matrix, $x_{id}$ denotes the $d$-th feature of node $i$, and $\mathcal{N}(i)$ represents the set of neighboring nodes of node $i$. This approach preserves the independence

of feature dimensions while emphasizing the relative importance of different neighbors in each dimension, thereby enhancing the model's ability to capture local structural information.

**Separable Attention.** In molecular systems, interatomic interactions are directional, and each atom plays two roles: target node and source node. Most existing methods only aggregate information from the source node to the target node (Liao & Smidt, 2023; Liao et al., 2024b; Wang et al., 2024; Shao et al., 2025), which assumes symmetric information flow. However, this is not always true in real physical systems. For example, in polar bonds, charged environments, or local defect structures, the effect of the source node on the target node can differ from the effect of the target node on the source node (Bengtsson, 1999; Kühne & Khaliullin, 2013). To address this, we introduce two independent sets of attention weights: source attention weights and target attention weights. The first models how neighboring nodes affect the central node, while the second captures how the central node influences its neighbors. In this way, the two roles of nodes are explicitly separated during aggregation. The overall workflow is illustrated in Figure 3(b). Given the interaction $e_{ij}$ between a target node $v_i$ and a source node $v_j$, we compute the attention weights as follows:

$$t_{ij} = \text{Linear}(e_{ij}) \quad \text{and} \quad ta_{ij} = \text{Dim-wise Softmax}(t_{ij}, \mathcal{N}(i)) \quad (2)$$

$$s_{ij} = \text{Linear}(e_{ij}) \quad \text{and} \quad sa_{ij} = \text{Dim-wise Softmax}(s_{ij}, \mathcal{N}(j)) \quad (3)$$

Here, $N(i)$ and $N(j)$ denote the indices of the target and source nodes, respectively. The final attention outputs are obtained as the weighted sum of $ta_{ij}$, $sa_{ij}$, and $e'_{ij}$. Here, $e'_{ij}$ is obtained by concatenating $e_{ij}$, $v_i$, and $v_j$, followed by target and source feature fusion through a gMLP (see Figure 3(d)). The two attention branches share the same computational flow and can therefore be executed in parallel. We also implement optimized kernels to improve training efficiency.

**Generality Analysis.** As mentioned earlier, many MLIPs are either unconstrained or equivariant. Unconstrained MLIPs are flexible, allowing Dim-wise Softmax and Separable Attention to be applied directly. For equivariant MLIPs, symmetry must be preserved. To ensure this, Dim-wise Softmax is computed on invariant features (e.g., $L = 0$), producing attention weights that are invariant. These weights are then applied to equivariant features within each irrep channel, without mixing components of different orders, ensuring the features remain equivariant under geometric transformations. Separable Attention extends this approach with two branches, computing attention equivariantly and aggregating information separately over the indices of the target and source nodes.

## 3.3 OVERALL ARCHITECTURE

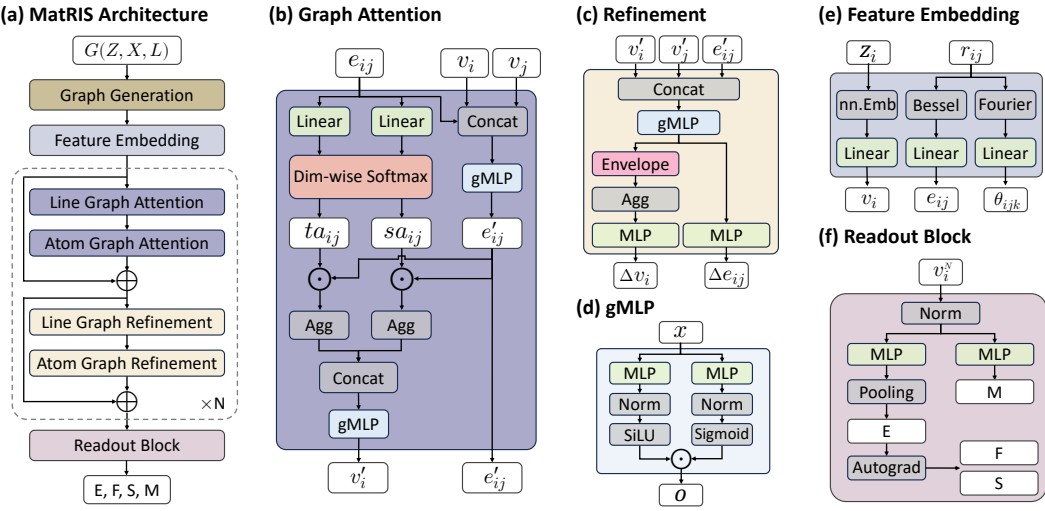

Figure 3: Overview of MatRIS. The model architecture (a) consists of feature embedding (e), graph attention (b), refinement (c), and a readout block(f).

**Graph Generation.** Given a crystalline system $G(Z, X, L)$, where $Z \in \mathbb{R}^{\text{natoms}}$ denotes the atomic numbers, $X \in \mathbb{R}^{\text{natoms} \times 3}$ represents the atomic coordinates, and $L \in \mathbb{R}^{3 \times 3}$ refers to the lattice. We first perform periodic repetition of the structure and then employ a radius-cutoff graph construction to represent it. Inspired by CHGNet (Deng et al., 2023), we construct the atom graph $G^a$ (with atoms as nodes and bonds as edges) based on $r_{cut}^a$ and the line graph $G^l$ (with bonds as nodes and angles as edges) based on $r_{cut}^l$ for the crystal structure.

**Feature Embedding.** Atomic numbers are initialized using trainable embeddings, while pairwise distances are encoded via a learnable radial Bessel basis with envelope functions (Gasteiger et al., 2020). For bond angles, we employ a learnable Fourier basis expansion following Deng et al. (2023). These input features are then passed through a linear layer to generate initial node, edge, and three-body representations. This block is illustrated in Figure 3(e).

**Refinement.** The output of Graph Attention $v_i'$ and $e_{ij}'$ is concatenated and processed by a gMLP operator (shown in Figure 3(c)). This representation is further refined by an Envelope function (Gasteiger et al., 2020) (ensures smoothness). The refined signals are then aggregated over neighboring interactions, enabling each edge to integrate local information. Finally, the output is transformed by a MLP operator to produce the residual node features $\Delta v_i$ and edge features $\Delta e_{ij}$.

**Readout Block.** The Readout Block (Figure 3(f)) takes node features $v_i$ from the last layer. After the normalization layer, the features are passed through MLPs to predict atomic energies and magnetic moments ($M$). Atomic energies are summed to obtain total energy ($E$). To ensure the reliability (Bigi et al., 2025), forces ($F_i$) and stress ($\sigma$) are computed by Equation 4. $\mathcal{V}$ is volume.

$$F_i = -\frac{\partial E}{\partial X_i}, \sigma = \frac{1}{\mathcal{V}} \frac{\partial E}{\partial \epsilon} \tag{4}$$

## 4 EXPERIMENTS

We evaluated MatRIS on Matbench-Discovery benchmark (Section 4.1), MatPES benchmark (Section 4.2), MDR phonon benchmark (Section 4.3), Molecular zero-shot benchmark (Section 4.4). The best results are in bold, and the second best are underlined. Moreover, we conducted ablation studies on the network modules and training methods (Section 4.5). Finally, we analyzed the efficiency of MatRIS (Section 4.6). In addition, the results of experiments on the Matbench-Discovery benchmark (non-compliant), LAMBench benchmark (Peng et al., 2026), Zeolite benchmark (Yin et al., 2025), DPA2 test sets (Zhang et al., 2024) and MD stability evaluation are provided in Appendix D. The training strategies are introduced in Appendix C, and the training hyperparameter settings are detailed in Appendix E.

### 4.1 MATBENCH DISCOVERY

**Dataset and Setting.** The Matbench-Discovery benchmark (Riebesell et al., 2025) is a well-established benchmark for evaluating the ability of models in new material discovery. In this benchmark, all models are required to optimize the geometry and predict the formation energy of each of the 256k structures in the WBM test set (Wang et al., 2021). These results are then used to assess thermodynamic stability at the ground state (0 K). In the "compliant" setting, all models are required to use MPTrj (Deng et al., 2023) as the training dataset, whereas in the "non-compliant" setting, this requirement is relaxed (the results are reported in Appendix D.1). More details on the hyperparameters can be found in Table 12. Moreover, inspired by Zaidi et al. (2023) and Liao et al. (2024a), we apply denoising pretraining to MatRIS-M and MatRIS-L. The specific details are described in Appendix E. Notably, this denoising is also adopted in several other works (ORB v2 MPTrj (Neumann et al., 2024), eqV2 S DeNS, and eSEN-30M-MP). Structures are relaxed using MatRIS and the FIRE (Bitzek et al., 2006) optimizer, with convergence reached after 500 steps or when the maximum force falls below 0.05 eV/Å.

**Results of the Compliant Benchmark** We summarize the comparison with other models in Table 1. MatRIS-S/M achieves performance comparable to eqV2 S DeNS/eSEN-30M-MP while using fewer parameters and lower computational costs (see Figure 1 for a comparison of training cost).

Table 1: MatRIS performance on the compliant Matbench-Discovery benchmark with results on unique structure prototypes. '↑'/'↓' stands for higher/lower is better.

| Model | Param. | F1↑ | DAF↑ | Precision↑ | Recall↑ | Accuracy↑ | MAE↓ | R2↑ | $K_{srme}$↓ | RMSD↓ |
|---|---|---|---|---|---|---|---|---|---|---|
| CHGNet | 0.41M | 0.613 | 3.361 | 0.514 | 0.758 | 0.851 | 0.063 | 0.689 | 1.717 | 0.0949 |
| MACE-MP-0 | 4.69M | 0.669 | 3.777 | 0.577 | 0.796 | 0.878 | 0.057 | 0.697 | 0.647 | 0.0915 |
| GRACE-2L | 15.3M | 0.691 | 4.163 | 0.636 | 0.757 | 0.896 | 0.052 | 0.741 | 0.525 | 0.0897 |
| Allegro-MP-L | 18.7M | 0.751 | 4.516 | 0.690 | 0.823 | 0.915 | 0.044 | 0.778 | 0.504 | 0.0816 |
| Nequix MP | 0.71M | 0.751 | 4.455 | 0.681 | 0.836 | 0.914 | 0.044 | 0.782 | 0.446 | 0.0853 |
| SevenNet-l3i5 | 1.17M | 0.760 | 4.629 | 0.708 | 0.821 | 0.920 | 0.044 | 0.776 | 0.550 | 0.0847 |
| NequIP-MP-L | 9.6M | 0.761 | 4.704 | 0.719 | 0.809 | 0.921 | 0.043 | 0.791 | 0.452 | 0.0856 |
| ORB v2 MPTrj | 25.2M | 0.765 | 4.702 | 0.719 | 0.817 | 0.922 | 0.045 | 0.756 | 1.725 | 0.1007 |
| HIENet | 7.51M | 0.777 | 4.932 | 0.754 | 0.801 | 0.929 | 0.041 | 0.793 | 0.642 | 0.0795 |
| Eqnorm MPTrj | 1.31M | 0.786 | 4.844 | 0.741 | 0.838 | 0.929 | 0.040 | 0.799 | 0.408 | 0.0837 |
| DPA-3.1-MPTrj | 4.81M | 0.803 | 5.024 | 0.768 | 0.841 | 0.936 | 0.037 | 0.812 | 0.650 | 0.0801 |
| eqV2 S DeNS | 31.2M | 0.815 | 5.042 | 0.771 | 0.864 | 0.941 | 0.036 | 0.788 | 1.676 | 0.0757 |
| eSEN-30M-MP | 30.1M | 0.831 | 5.260 | 0.804 | 0.861 | 0.946 | 0.033 | 0.822 | **0.340** | 0.0752 |
| MatRIS-S | 4.3M | 0.811 | 5.127 | 0.784 | 0.840 | 0.940 | 0.036 | 0.803 | 0.730 | 0.0766 |
| MatRIS-M | 6.3M | 0.833 | 5.363 | 0.820 | 0.847 | 0.948 | 0.033 | 0.820 | 0.542 | 0.0742 |
| MatRIS-L | 10.4M | **0.847** | **5.422** | **0.829** | **0.865** | **0.951** | **0.031** | **0.829** | 0.489 | **0.0717** |

These results demonstrate the effectiveness of MatRIS. Moreover, MatRIS-L achieves SOTA performance across all metrics, with an F1 score of 0.847. It also achieves an RMSD of 0.0717 when comparing relaxed structures to DFT reference values.

## 4.2 MATCALC BENCHMARK

Table 2: Summary of model performance on the MatCalc benchmark. All results were obtained using the `MatCalc` package and its associated dataset.

| Model | Param. | Equilibrium | | Near-equilibrium | | | |
|---|---|---|---|---|---|---|---|
| | | $d$↓ | $Ef$↓ | $K$↓ | $G$↓ | $CV$↓ | $f/f_{DFT}$↑ |
| MatPES-trained models | | | | | | | |
| M3GNet | 0.66M | 0.42 | 0.11 | 26 | 25 | 27 | **0.97** |
| CHGNet | 2.7M | 0.43 | 0.082 | 24 | 21 | 23 | 0.91 |
| TensorNet | 0.84M | **0.37** | 0.081 | 18 | 15 | **13** | 0.93 |
| MatRIS | 1.4M | 0.54 | **0.068** | **15** | **13** | 16 | 0.96 |
| MPTrj-trained models | | | | | | | |
| CHGNet | 2.7M | 0.51 | 0.092 | 17.0 | 30.0 | 24.0 | 0.830 |
| MACE-L | 5.7M | 0.43 | – | 25.3 | 22.5 | 11.6 | 0.829 |
| SevenNet-l3i5 | 1.17M | 0.55 | 0.057 | 13.2 | 170.2 | 8.03 | 0.922 |
| eqV2 S DeNS | 31.2M | **0.25** | **0.033** | 27.2 | 29.1 | 25.9 | 0.964 |
| eSEN-30M-MP | 30.1M | 0.34 | 0.039 | 18.8 | 77.3 | **4.66** | **0.986** |
| MatRIS-M | 6.3M | 0.32 | 0.041 | **12.4** | **16.4** | 7.39 | 0.983 |
| OAM-trained models | | | | | | | |
| SevenNet-MF-ompa | 25.7M | 0.502 | 0.028 | 13.3 | 32.2 | 4.60 | 0.976 |
| eqV2 M | 86.6M | **0.235** | **0.017** | 25.4 | 17.5 | 80.4 | **0.999** |
| eSEN-30M-OAM | 30.1M | 0.299 | 0.089 | 11.9 | 14.8 | 4.35 | 0.996 |
| MatRIS-10M-OAM | 10.4M | 0.316 | 0.025 | **10.6** | **13.3** | **3.97** | 0.985 |

**Dataset and Settings.** MatCalc benchmark (Kaplan et al., 2025) covers equilibrium properties (relaxed structure similarity $d$, formation energy $Ef$), near-equilibrium properties (bulk modulus $K$, shear modulus $G$, constant-volume heat capacity $CV$, force softening $f/f_{DFT}$), and is constructed from test data collected from the Materials Project (Jain et al., 2013a), Alexandria (Schmidt et al., 2024), WBM high-energy states (Wang et al., 2021). We compare models trained on MatPES-PBE (Kaplan et al., 2025), MPTrj and OAM, with the results reported in Table 2. We maintained consistent evaluation settings for all MPTrj- and OAM-trained models. Specifically, for elastic moduli calculations, we applied normal (diagonal) strain magnitudes of $\pm 0.01$ and $\pm 0.005$ for $\epsilon_{11}, \epsilon_{22}, \epsilon_{33}$, and off-diagonal strain magnitudes of $\pm 0.06$ and $\pm 0.03$ for $\epsilon_{23}, \epsilon_{13}, \epsilon_{12}$.

**Results and Analysis.** Table 2 summarizes the performance of models trained on the MatPES-PBE, MPTrj and OAM datasets. For models trained on MatPES-PBE, MatRIS achieves the best overall performance, clearly outperforming other models in predicting formation energy $Ef$.

For models trained on MPTrj, MatRIS attains SOTA or near-SOTA results on 83% of the evaluated metrics. Notably, MatRIS remains robust on "Near-equilibrium" tasks regardless of the training dataset. Although the higher fraction of near-equilibrium structures in MPTrj amplifies PES "Softening" ($f/f_{DFT}$) (Deng et al., 2024; Kaplan et al., 2025), MatRIS maintains stable performance. Comparisons with CHGNet further indicate that changing the training dataset does not result in significant softening, underscoring the effectiveness of the architectural design.

The same trend is observed for models trained on OAM, where MatRIS-10M-OAM performs robustly across all metrics and achieves SOTA performance overall.

## 4.3 MDR PHONON BENCHMARK

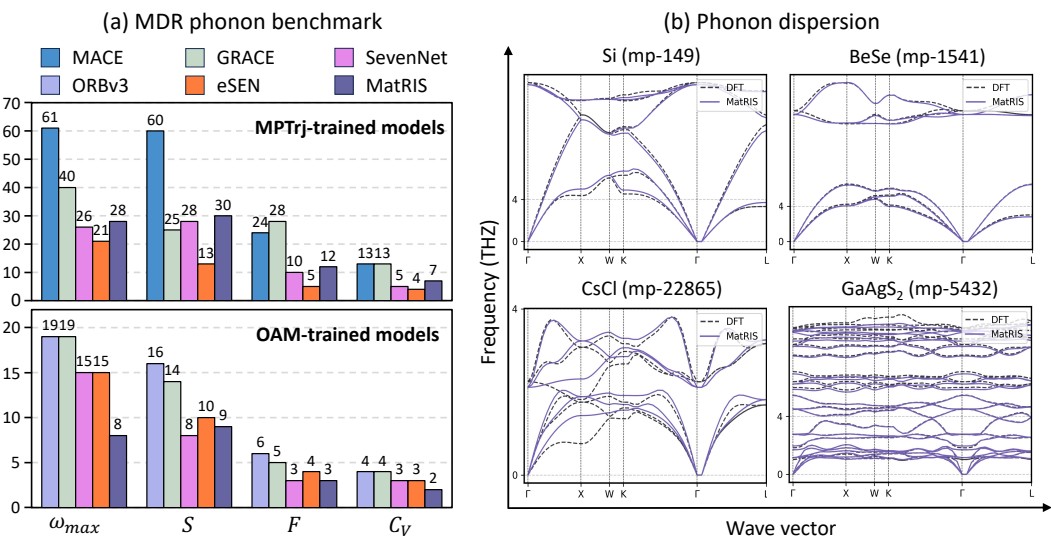

Figure 4: (a) Summary of model performance on the MDR phonon benchmark. The evaluation metrics include $\omega_{max}$ (K), $S$ (J/K/mol), $F$ (kJ/mol), and $C_V$ (J/K/mol), where the reported values represent the MAE between the model predictions and the DFT results. (b) Predicted phonon dispersion obtained using MatRIS with a 0.01Å displacement. The DFT results are taken from the phononDB dataset.

**Dataset and Settings.** The MDR phonon benchmark (Loew et al., 2025) is used to evaluate the ability of MLIPs to predict phonon properties. It requires MLIPs to compute the maximum phonon frequency ($\omega_{max}$), entropy ($S$), free energy ($F$), and heat capacity at constant volume ($C_V$) for approximately 10,000 structures. To ensure a fair comparison, we follow the evaluation protocol adopted in Loew et al. (2025). Specifically, we first optimize the structures using the FIRE optimizer (max steps=500, fmax=0.005). Displacements are generated with a magnitude of 0.01 Å, and the properties are computed at 300 K.

**Results and Analysis.** Results for both the MPTrj-trained and OAM-trained models are shown in Figure 4(a). MatRIS achieves competitive results among the MPTrj-trained models. As the training dataset grows, MatRIS-10M-OAM achieves SOTA accuracy on most metrics, with a particularly significant improvement in the maximum phonon frequency ($\omega_{max}$). It is worth noting that MatRIS-10M-OMat achieves higher accuracy, with values of $\omega_{max}$, $S$, $F$, and $C_V$ being 7.08, 7.12, 2.12, and 1.91, respectively.

We also selected four representative structures from the phononDB dataset (Togo) for visualization, as shown in Figure 4(b). We find that MatRIS-MP-L not only reproduces the highest frequencies but also aligns well with the DFT results overall.

## 4.4 MOLECULAR ZERO-SHOT BENCHMARK

Table 3: MatRIS performance on the Molecular zero-shot benchmark. Energy (E) is reported in kcal/mol and Force (F) in kcal/mol/Å.

| Model | TorsionNet-500 | | | | MD22 (Ac-Ala3-NHMe) | | ANI-1x | | AIMD-Chig | |
|---|---|---|---|---|---|---|---|---|---|---|
| | E MAE | E RMSE | MAEB[a] | NABH$_h^b$ | E | F | E | F | E | F |
| AIMNet2 | 0.38 | 0.55 | 0.58 | 82 | – | – | – | – | – | – |
| DPA2-Drug | 0.24 | 0.35 | 0.36 | 18 | – | – | – | – | – | – |
| MACE-OFF-L | 0.14 | 0.21 | 0.23 | 4 | 2.29 | 5.205 | 5.82 | **3.948** | 8.25 | 3.853 |
| DPA3-L24 | 0.06 | 0.09 | 0.09 | **0** | – | – | – | – | – | – |
| MatRIS-M | **0.04** | **0.07** | **0.07** | **0** | **2.23** | **5.140** | **4.15** | 4.414 | **1.55** | **3.604** |

[a] The MAE of the torsional barrier height is defined as the energy difference between the minimum and maximum along the torsional rotation.
[b] The number of molecules with barrier height errors exceeding 1 kcal/mol.

**Dataset and Settings.** AIMNet and DPA2-Drug were trained on the datasets from Anstine et al. (2025) and Yang et al. (2025), respectively, while the other models were pre-trained on SPICE-MACE-OFF23 (Kovács et al., 2025), which contains 951,005 small-molecule configurations. The energies and forces of all configurations were computed at the $\omega$B97M-D3(BJ)/def2-TZVPPD level. Following Peng et al. (2026), we selected the Ac-Ala3-NHMe from the MD22 (Chmiela et al., 2023), ANI-1x (Smith et al., 2018), and AIMD-Chig (Wang et al., 2023) datasets, which contain 2,212, 8,861, and 19,800 configurations, respectively. The training parameters can be found at Table 12 in appendix. For the MD22, ANI-1x, and AIMD-Chig benchmarks, we corrected inconsistencies caused by differences between DFT functionals.

**Results and Analysis.** We selected available molecular pre-trained models (MACE-OFF-L (Kovács et al., 2025) and DPA3-L24 for comparison, and the results are summarized in Table 3. MatRIS-M performs well across four downstream datasets. On the TorsionNet-500 test set, its energy prediction error is reduced by about 22.2%–33.3% compared to the current SOTA model DPA3. In the remaining three datasets, MatRIS-M ranks first in five out of six metrics and second in the remaining one, demonstrating that it effectively leverages the knowledge from the pre-training dataset to achieve more reliable performance on downstream tasks.

## 4.5 ABLATION STUDY

We conduct ablation studies on various modules of MatRIS and their corresponding training methods. We train the MatRIS-S on MPTrj and evaluate it on 15,000 randomly sampled structures from the WBM dataset, reporting performance using the MAE of formation energies.

Table 4: Ablation studies. Formation energy (**Ef**) MAE is in meV/atom, lower is better.

| Index | Dim-wise softmax | Separable attention | Learnable envelope | Ef (MAE) |
|---|---|---|---|---|
| 1 | ✓ | ✓ | ✓ | 28.0 |
| 2 | ✗ | ✓ | ✓ | 28.4 |
| 3 | ✗ | ✗ | ✓ | 29.1 |
| 4 | ✗ | ✗ | ✗ | 31.3 |

(a) Effect of modules: dim-wise softmax, separable attention and learnable envelope.

| Index | Denoising pretraining | With magmom | Graph-level loss | Ef (MAE) |
|---|---|---|---|---|
| 1 | ✓ | ✓ | ✓ | 27.2 |
| 2 | ✗ | ✓ | ✓ | 28.0 |
| 3 | ✗ | ✗ | ✓ | 29.7 |
| 4 | ✗ | ✗ | ✗ | 30.2 |

(b) Effect of training methods: denoising pretraining, magnetic moment prediction, and graph-level loss.

**Module Ablation.** We evaluated the impact of Dim-wise Softmax, Separable Attention, and Learnable Envelope on model accuracy, as shown in Table 4(a). Specifically, replacing Dim-wise Softmax with standard softmax (i.e., sharing the same weight across all feature dimensions) increased MAE to 28.4 meV/atom (Index 2). Subsequently, restricting attention-based aggregation to the source-to-target direction further increased MAE (Index 3). Finally, replacing the learnable envelope function with a fixed one also degraded performance (Index 4).

**Training Method Ablation.** We further analyze the contributions of different training strategies to model performance, as shown in Table 4(b). Pretraining with denoising (see Section C.3 in appendix) effectively improves performance, as predicting noise helps mitigate over-smoothing (Godwin et al., 2022; Zaidi et al., 2023). Notably, predicting magmoms also enhances accuracy, since magmoms help distinguish features in different chemical environments and, being a node-level task, likely reduce over-smoothing during training (Deng et al., 2023). Finally, performing loss reduction at the graph level prevents the force loss from being biased by differences in system size (see Section C.2 in appendix for more detail), further improving performance.

## 4.6 EFFICIENCY-ACCURACY ANALYSIS.

In this section, we compare the computational efficiency of MatRIS with several SOTA models (benchmarked without `torch.compile` or optimized kernels).

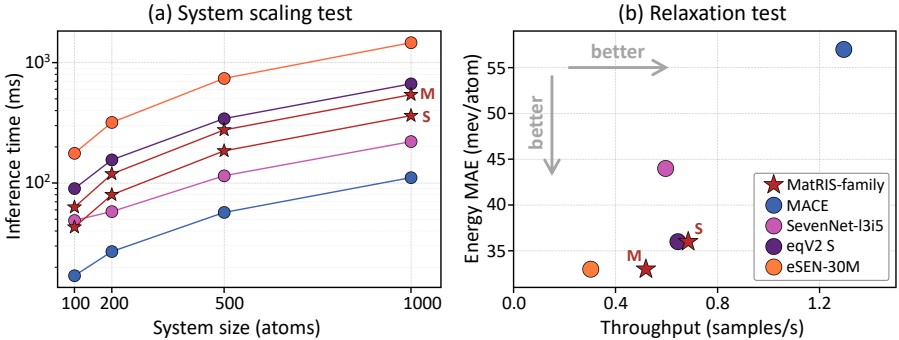

Figure 5: Efficiency-accuracy comparison.

The inference time across system sizes (excluding graph construction) is summarized in Figure 5(a). MatRIS demonstrates higher efficiency than eqV2 and eSEN but lower than MACE-L, as MACE-L utilizes only 2 layers while MatRIS-S and MatRIS-M employ 4 and 6 layers, respectively.

We further evaluate the models in practical applications. We randomly select 500 structures from the WBM dataset (Wang et al., 2021) and perform relaxations using ASE, recording the throughput and energy MAE for each model. As shown in Figure 5(b), MatRIS achieves a favorable balance between speed and accuracy, maintaining high precision while enabling faster computation.

## 5 CONCLUSION AND FUTURE WORK

In this work, we review the characteristics of invariant, equivariant and unconstrained MLIPs in the era of rapidly expanding QM-based datasets. Equivariant MLIPs deliver superior accuracy while their reliance on tensor products and high-degree representations leads to prohibitive computational costs. Motivated by the question of whether strict equivariance remains indispensable, we introduce MatRIS, an invariant MLIP that leverages attention to model three-body interactions. Across multiple benchmarks, MatRIS attains SOTA results, opening a new path toward accurate and efficient MLIPs.

In future work, we will scale MatRIS to even larger QM-based datasets to further validate its expressiveness. We aim to develop a reliable distillation strategy to develop student MLIPs. We also plan to expand MatRIS to incorporate long-range electrostatics, thereby enhancing its applicability to more complex downstream tasks.

ACKNOWLEDGMENTS

This work is supported by the following funding: National Science Foundation of China (T2125013, 92270206, 62372435), the Innovation Funding of ICT, CAS. Part of the training is performed on the robotic AI-Scientist platform of Chinese Academy of Sciences. The authors thank the ICT operations team for their strong support.

We thank Duo Zhang for helpful discussions. We also thank Lijun Liu (Osaka University) for evaluating the eqV2 Liao et al. (2024b); Barroso-Luque et al. (2024) and eSEN Fu et al. (2025).

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

APPENDIX

# A  ADDITIONAL RELATED WORKS

## A.1  EXPLICIT THREE-BODY MODELING IN MLIPS

Three-body information is important for predicting various material properties (Choudhary & De-Cost, 2021; Choudhary et al., 2023; Zhang et al., 2025). Several MLIPs explicitly model three-body interactions, including the ALIGNN series (Choudhary & DeCost, 2021; Choudhary et al., 2023), DimeNet series (Gasteiger et al., 2020; 2022a), GemNet series (Gasteiger et al., 2022b; Klicpera et al., 2021), M3GNet series (Chen & Ong, 2022; Yang et al., 2024), and DPA3 (Zhang et al., 2025), where three-body features are updated within the interaction block and then used to refine edge and node features. MatRIS also explicitly models three-body interactions, but differs in its update and aggregation strategy:

1. In MatRIS, three-body features are updated using an attention mechanism with $O(N)$ complexity, which improves model accuracy while maintaining computational efficiency (see Table 4(a), Index 1 and 3).
2. During message aggregation, a learnable envelope function is used to smooth the three-body contributions, instead of using a simple distance-based attenuation. This design yields better performance (see Table 4(a), Index 3 and 4).

Beyond the above distinctions, MatRIS also achieves SOTA or near-SOTA performance across multiple benchmarks. Notably, on Matbench-Discovery (Riebesell et al., 2025), MatRIS-M attains accuracy comparable to eSEN-30M-MP but with significantly fewer parameters and substantially lower

training cost. This demonstrates the effectiveness of combining three-body interaction modeling with attention mechanisms for materials modeling.

## A.2 ATTENTION-BASED MLIPs

**Full Attention MLIPs.** In potential energy surface modeling, full-attention methods have been widely used, with representative models including DPA2 (Zhang et al., 2024), PET (Pozdnyakov & Ceriotti, 2023), and EScAIP (Qu & Krishnapriyan, 2024). In these models, all features are concatenated into a message tensor of shape $[\text{natoms}, \text{neighbors}, D]$, where $D$ is the hidden dimension, and updated using a full attention mechanism (Vaswani et al., 2023). This mechanism can fully leverage information from neighboring atoms and enable feature interactions. However, due to the $O(N^2)$ computational cost of full attention, the computation quickly becomes expensive as the system size grows. To maintain efficiency, these MLIPs typically consider only two-body interactions.

In contrast, MatRIS employs an attention mechanism with $O(N)$ complexity to capture both two-body and three-body interactions. This allows MatRIS to incorporate more geometric information and enhance the model's expressive power while maintaining computational efficiency.

**Linear-Complexity Attention MLIPs.** Although there exist many MLIPs with attention mechanisms of $O(N)$ complexity, such as Equiformer (Liao & Smidt, 2023), ViSNet (Wang et al., 2024), FreeCG (Shao et al., 2025), and MGT (Anselmi et al., 2024), we note that these architectures still differ from MatRIS in several technical aspects.

First, Equiformer is a high-degree steerable equivariant MLIP that employs multi-layer perceptron attention (Liao & Smidt, 2023; Brody et al., 2022) for feature encoding, with $O(N)$ complexity, and can support vectors of any degree $L$. However, unlike MatRIS, its attention mechanism is not directly used to encode three-body interactions and only considers the source-to-target direction.

ViSNet and FreeCG belong to equivariant MLIPs. Their attention mechanisms can capture up to four-body interactions and also have $O(N)$ complexity. Nevertheless, their technical approaches differ fundamentally from that of MatRIS: ViSNet and FreeCG implicitly extract and refine three- or four-body features, without directly employing attention to encode and update such features. In contrast, MatRIS explicitly extracts three-body features and utilizes attention mechanisms to encode and update them.

It is also worth noting that MGT employs attention and explicitly extracts three-body features. Nevertheless, its attention mechanism operates on node and edge feature updates, and relies on the ALIGNN module (Choudhary & DeCost, 2021) to further refine three-body features. Similar to ViSNet and FreeCG, MGT leverages attention to refine three-body features rather than directly encode them.

In contrast to all the above attention-based models, MatRIS introduces a separable attention mechanism that explicitly models three-body interactions. Its key innovations include:

1. **Dimension-wise**: the attention weights vary across feature dimensions, distinguishing the relative importance of different dimensions.

2. **Separable**: it considers both source-to-target and target-to-source directions, generating separate attention weights for each direction.

3. **Explicit and efficient modeling**: due to its $O(N)$ complexity, MatRIS can efficiently and explicitly encode and update three-body features, thereby significantly enhancing the model's expressive power.

## B DETAILS OF MATRIS

For completeness, in this section we present the details of MatRIS. Given a material graph $G = (Z, X, L)$, the atomic numbers are denoted as $Z \in \mathbb{R}^{\text{natoms} \times 1}$, the atomic positions as $X \in \mathbb{R}^{\text{natoms} \times 3}$, and the lattice as $L \in \mathbb{R}^{3 \times 3}$. In **Graph Generation** (see Figure 6), we first perform periodic repetition, and then construct the Atom Graph $G^a$ and the Line Graph $G^l$ based on radial cutoffs $r_{cut}^a$ and $r_{cut}^l$, respectively. We typically set $r_{cut}^l < r_{cut}^a$ for computational efficiency.

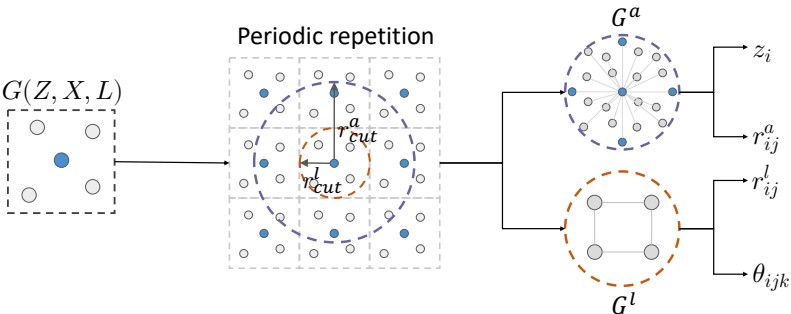

Figure 6: The detail of Graph Generation.

In **Feature Embedding**, the atomic types $z_i$ are expanded using a learnable embedding matrix $A \in \mathbb{R}^{\text{max\_element} \times D}$ (see Equation 5). Following Gasteiger et al. (2020), the pairwise distances $r_{ij}^a$ of $G^a$ are expanded with Bessel basis functions, and smoothed with a envelope function($\mu(r_{ij})$) to ensure that pairwise interactions decay smoothly to zero at the cutoff radius, as shown in Equation 6. For the initialization of three-body interactions $\theta_{ijk}$, we follow Deng et al. (2023) and employ the computationally simpler Fourier basis instead of spherical harmonics (we also tested spherical harmonics but observed no change in accuracy), as shown in Equation 7. Finally, all initialized features undergo nonlinear transformations and are projected into a higher-dimensional space.

$$v_i^0 = W(z_i A) + b \tag{5}$$

$$e_{ij}^0 = \mu(r_{ij}^a) \cdot \sqrt{\frac{2}{r_a}} \frac{\sin\left(\frac{n\pi}{r_a} r_{ij}^a\right)}{r_{ij}^a} \tag{6}$$

$$a_{ijk,m}^0 = \begin{cases} \frac{1}{\sqrt{\pi}} \cos[m\theta_{ijk}], & m \in [0, M_{max}/2], \\ \frac{1}{\sqrt{\pi}} \sin[(m - M_{max}/2)\theta_{ijk}], & m \in [M_{max}/2 + 1, M_{max}]. \end{cases} \tag{7}$$

In the **Graph Attention** module, we consider the $n$-th layer, which contains the Atom Graph $G^a$ with features $v_i^n$ and $e_{ij}^n$, and the Line Graph $G^l$ with features $e_{ij}^n$ and $\theta_{ijk}^n$. The two graphs share the edge features $e_{ij}^n$. To fuse information between the graphs, we first update $G^l$ and then incorporate its output into $G^a$ to aid atomic feature updates (see Equation 8). Specifically, $e_{ij}^n$ and $\theta_{ijk}^n$ are input to the 'Line Graph Attention' module, producing $e_{ij,l}^{'(n)}$ and $\theta_{ijk}^{'(n)}$. The updated edge feature $e_{ij}^{'(n)}$ encodes three-body interactions and is combined with $v_i^n$ as input to the 'Atom Graph Attention' module, yielding $v_i^{'(n)}$ and $e_{ij}^{'(n)}$.

$$\begin{aligned} e_{ij,l}^{'(n)}, \theta_{ijk}^{'(n)} &\leftarrow \text{Line Graph Attention}(e_{ij}^n, \theta_{ijk}^n) \\ v_i^{'(n)}, e_{ij}^{'(n)} &\leftarrow \text{Atom Graph Attention}(v_i^n, e_{ij,l}^{'(n)}) \end{aligned} \tag{8}$$

Notably, in $G^a$, atoms are treated as nodes and pairwise distances as edges, whereas in $G^l$, pairwise distances are treated as nodes and three-body interactions as edges. Therefore, the computation of the Line Graph Attention and Atom Graph Attention follows the same computaion, differing only in their inputs. Here, we take the update of $G^a$ as an example and present the detailed operations of the Graph Attention module.

Given the input of the current attention layer, $v_i$ and $e_{ij}$ (for simplicity, we denote $v_i^n$ and $e^{'(n)}ij, l$ as $v_i$ and $eij$), we first generate the fusion feature:

$$e_{ij}' = \text{gMLP}(v_i||v_j||e_{ij}) \tag{9}$$

Meanwhile, $e_{ij}$ undergoes two nonlinear transformations. It is then passed through Dim-wise Softmax to get the attention weights for the source and target nodes, $sa_{ij}$ and $ta_{ij}$. These weights are

then element-wise multiplied with $e'_{ij}$ and fused to produce the output of the attention layer, $v'_i$, as shown in the equation:

$$tv_i = \sum_{k \in \mathcal{N}(i)} \left( ta_{kj} \odot e'_{kj} \right), \quad sv_i = \sum_{k \in \mathcal{N}(j)} \left( sa_{kj} \odot e'_{kj} \right)$$

$$v'_i = \text{gMLP}\left(tv_i \| sv_i\right)$$

(10)

After the Attention layer, we employ a **Refinement** layer to further enhance the geometric encodings. In this module, we also apply a learnable envelope function to smooth the potential energy surface. As before, we first update $G^l$ and then $G^a$. Taking the update of $G^a$ as an example, the inputs are $v'_i$ and $e'_{ij}$, and we fuse the features from the attention layer:

$$m_{ij}^{(n)} = \text{gMLP}(v'_i \| v'_j \| e'_{ij})$$

(11)

The fusion feature $m_{ij}^{(n)}$ is transformed nonlinearly to yield the edge update $\Delta e_{ij}$. For the node update $\Delta v_i$, we first apply the learnable envelope function (Equation 12), then aggregate, and finally apply a nonlinear transformation.

$$\mu_{ij}^{(n)} = \text{Linear}(e_{ij}^0)$$

(12)

$$\Delta v_i = \text{MLP}\left( \sum_{k \in \mathcal{N}(i)} (\mu_{ij}^{(n)} \odot m_{ij}^{(n)}) \right)$$

$$\Delta e_{ij} = \text{MLP}(m_{ij}^{(n)})$$

(13)

In the Readout block, the final atom features $v_i^{(N)}$ are used to directly predict the total energy ($E$) and magnetic moments ($M$):

$$E = \sum_i \text{MLP}(v_i^N) + \text{ref}(z_i)$$

$$M = \text{MLP}(v_i^N)$$

(14)

Here, $\text{ref}(z_i)$ is obtained by performing a least-squares fit to the dataset energies. The atomic forces $F_i$ and stress ($\sigma$) are obtained via automatic differentiation of the energy with respect to the atomic Cartesian coordinates $X_i$ and the lattice strain tensor ($\varepsilon$).

$$F_i = -\frac{\partial E}{\partial X_i}, \quad \sigma = \frac{1}{\mathcal{V}} \frac{\partial E}{\partial \epsilon}$$

(15)

## C  TRAINING STRATEGIES

### C.1  LOAD-BALANCE STRATEGY

In most atomic datasets, the sizes of structures typically follow a long-tailed distribution (Deng et al., 2023; Barroso-Luque et al., 2024). Figure 7(a) visualizes the distributions of atom and edge counts in the MPTrj dataset (cutoff radius 6.0). In distributed training, allocating samples with a fixed batch size may cause two issues: (1) a GPU may receive only large samples, leading to out-of-memory (OOM) and limiting the maximum batch size; (2) assuming memory allows, some GPUs may have higher computational loads while others have lower loads, causing idle time and extra synchronization overhead. To address this, we adopt a **load balancing strategy**:

1. Shuffle the entire dataset to ensure randomness, then split it into multiple chunks.

2. Within each chunk, sort samples in descending order of size.

3. Using a greedy algorithm, assign samples sequentially to GPUs, prioritizing the GPU with the largest remaining capacity. If all GPUs are "full" or the chunk has no remaining samples, a batch is generated.

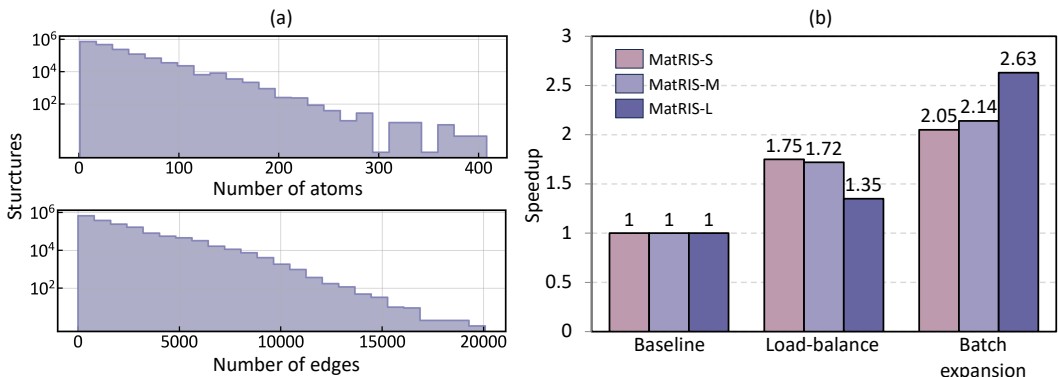

Figure 7: (a)Number of atoms and edges in MPTrj dataset. (b)Training speedup achieved by load balancing.

As shown in Figure 7(b), with load balancing, the cumulative speedup reaches **1.35–1.75×**, indicating that synchronization overhead due to load imbalance is significant during each training step. Building on this, the global batch size can be increased (denoted as "Batch expansion"), resulting in a cumulative speedup of **2.05–2.63×**.

## C.2 LOSS-BALANCE STRATEGY

Besides load imbalance during training, the loss computation can also be imbalanced. This is especially clear for the force and magmom loss(e.g., node-level tasks). For example, the MSE loss (forces only) is defined as:

$$L_f = \frac{1}{3N_B} \sum_{i=1}^{N_B} \left| f_i^{\text{pred}} - f_i^{\text{DFT}} \right|^2 \tag{16}$$

where $N_B$ denotes the total number of atoms in this global batch.

This reduction method causes the losses of large samples to dominate during training, effectively making the model prioritize large samples and resulting in a loss imbalance. To address this, we adopt a graph-level loss reduction strategy (referred to as **Graph-level loss**). Specifically, when computing node-level losses such as force or magmom, we first reduce the loss within each graph, and then perform a second reduction across the global batch, thereby mitigating the loss imbalance. Ablation studies in Table 4(b) demonstrate the effectiveness of this method. The corresponding formula is as follows:

$$L_f = \frac{1}{B} \sum_{b=1}^{B} \left( \frac{1}{N_b} \sum_{i=1}^{N_b} \left\| f_i^{\text{pred}} - f_i^{\text{DFT}} \right\|^2 \right) \tag{17}$$

where $B$ is the number of graphs in this global batch, $N_b$ is the number of atoms in the $b$-th graph.

## C.3 DETAILS OF DENOISING PRETRAINING

In this section, we present the denoising strategy used in our work. Several studies, such as Noisy Node (Godwin et al., 2022) and DeNS (Liao et al., 2024a), have demonstrated that denoising can mitigate the over-smoothing of GNNs and improve generalization. However, these methods have restricted applicability. For example, Noisy Node must be applied to equilibrium structures, while DeNS extends it to non-equilibrium states but is limited to equivariant neural networks. In practice, most mainstream atomic datasets are not fully in equilibrium (Deng et al., 2023; Schmidt et al., 2024; Barroso-Luque et al., 2024), and MLIPs are not necessarily equivariant GNNs. The training of MatRIS faces this scenario. For non-equivariant GNNs on non-equilibrium structures, a key challenge in applying denoising is how to encode forces. Inspired by Liao et al. (2024a), we project forces onto edges to scalarize the vectors. The detailed denoising strategy is described step by step as follows:

1. Given a corruption probability, we randomly select atoms to corrupt.

2. For each corrupted atom, we project its force onto the relative position edges by computing inner products, resulting in a scalarized force feature $F$(projected).

3. We sample a random timestep $t$ and obtain the noise standard deviation $\sigma_t$ from a linear schedule. We then add noise as $\sigma_t \epsilon$, where $\epsilon$ is standard Gaussian noise.

4. The noisy structure, the projected force $F$(projected), and the timestep $t$ are used as inputs to the MatRIS model, which directly predicts the added Gaussian noise.

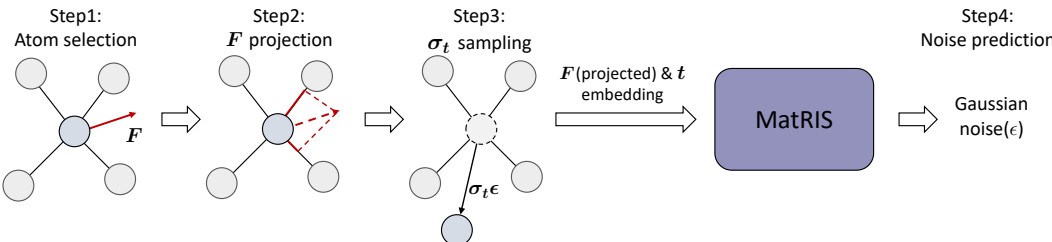

Figure 8: The workflow of denoising pretraining on MatRIS.

The overall workflow is illustrated in Figure 8. After the denoising pretraining stage, the model is further fine-tuned using the pretrained weights.

# D  ADDITIONAL EVALUATION

## D.1  MATBENCH-DISCOVERY BENCHMARK (NON-COMPLIANT)

We report the performance of non-compliant MatRIS-10M-OAM models, all of which were accessed before September 24, 2025. In this benchmark, the training data for each model is not restricted. We follow the setup used by most models, with pretraining on the OMat24 (Barroso-Luque et al., 2024) dataset and joint fine-tuning with sAlex (Schmidt et al., 2024; Barroso-Luque et al., 2024) and MPTrj (Deng et al., 2023). Results on the full, unique, and 10k most stable splits are shown in Table 5, Table 6, and Table 7, respectively. Although MatRIS-10M-OAM has fewer parameters than most models, it still delivers competitive performance across all splits.

Table 5: MatRIS performance on the non-compliant Matbench-Discovery benchmark with results on **full test set**.

| Model | Param. | F1↑ | DAF↑ | Precision↑ | Recall↑ | Accuracy↑ | MAE↓ | R2↑ | RMSD↓ |
|---|---|---|---|---|---|---|---|---|---|
| GRACE-2L-OAM-L | 26.4M | 0.862 | 5.093 | 0.874 | 0.851 | 0.953 | 0.022 | 0.856 | 0.064 |
| DPA-3-3M-FT | 3.27M | 0.864 | 4.912 | 0.843 | 0.887 | 0.952 | 0.022 | _0.862_ | 0.069 |
| Nequip-OAM-L | 9.6M | 0.870 | 5.060 | 0.868 | 0.872 | 0.955 | 0.022 | 0.858 | 0.065 |
| Allegro-OAM-L | 9.7M | 0.873 | 4.876 | 0.837 | **0.912** | 0.954 | 0.021 | 0.861 | 0.065 |
| AlphaNet-v1-OMA | 4.65M | 0.883 | 5.000 | 0.858 | _0.910_ | 0.959 | 0.023 | 0.827 | 0.079 |
| SevenNet-MF-ompa | 25.7M | 0.884 | 5.082 | 0.872 | 0.895 | 0.960 | 0.021 | 0.861 | 0.064 |
| ORB v3 | 25.5M | 0.887 | 5.159 | 0.885 | 0.888 | 0.961 | 0.023 | 0.820 | 0.075 |
| eqV2 M | 86.6M | 0.896 | 5.243 | 0.900 | 0.893 | _0.965_ | 0.020 | 0.842 | 0.069 |
| eSEN-30M-OAM | 30.2M | _0.902_ | **5.281** | **0.906** | 0.899 | **0.967** | **0.018** | 0.860 | _0.061_ |
| MatRIS-10M-OAM | 10.4M | **0.903** | _5.275_ | _0.905_ | 0.901 | **0.967** | _0.019_ | **0.864** | **0.060** |

## D.2  LAMBENCH BENCHMARK

We evaluated the performance of MatRIS-10M-OAM on LAMBench benchmark (Peng et al., 2026), which mainly tests the generalizability of MLIPs. The benchmark covers test domains including molecules, inorganic materials, and catalysis, and includes tasks such as force field prediction and property calculation.

Table 6: MatRIS performance on the non-compliant Matbench-Discovery benchmark with results on **unique structure prototypes**.

| Model | Param. | F1↑ | DAF↑ | Precision↑ | Recall↑ | Accuracy↑ | MAE↓ | R2↑ | $K_{srme}$↓ | RMSD↓ |
|---|---|---|---|---|---|---|---|---|---|---|
| GRACE-2L-OAM-L | 26.4M | 0.883 | 5.840 | 0.893 | 0.874 | 0.964 | 0.022 | 0.862 | 0.169 | 0.064 |
| DPA3-3M-FT | 3.27M | 0.884 | 5.667 | 0.866 | 0.903 | 0.963 | 0.023 | 0.869 | 0.469 | 0.069 |
| Nequip-OAM-L | 9.6M | 0.893 | 5.832 | 0.890 | 0.895 | 0.967 | 0.022 | 0.865 | **0.166** | 0.065 |
| Allegro-OAM-L | 9.7M | 0.895 | 5.674 | 0.867 | 0.923 | 0.966 | 0.022 | 0.868 | 0.319 | 0.065 |
| AlphaNet-v1-OMA | 4.65M | 0.901 | 5.747 | 0.879 | 0.924 | 0.968 | 0.024 | 0.831 | 0.644 | 0.079 |
| SevenNet-MF-ompa | 25.7M | 0.901 | 5.825 | 0.890 | 0.911 | 0.969 | 0.021 | 0.867 | 0.317 | 0.064 |
| ORB v3 | 25.5M | 0.905 | 5.912 | 0.904 | 0.907 | 0.971 | 0.024 | 0.821 | 0.210 | 0.075 |
| eqV2 M | 86.6M | 0.917 | 6.047 | 0.924 | 0.910 | 0.975 | 0.020 | 0.848 | 1.771 | 0.069 |
| eSEN-30M-OAM | 30.2M | **0.925** | **6.069** | **0.928** | **0.923** | **0.977** | **0.018** | 0.866 | 0.170 | 0.061 |
| MatRIS-10M-OAM | 10.4M | 0.921 | 6.039 | 0.923 | 0.918 | 0.976 | 0.019 | **0.871** | 0.218 | **0.060** |

Table 7: MatRIS performance on the non-compliant Matbench-Discovery benchmark with results on **10k most stable**.

| Model | Param. | F1↑ | DAF↑ | Precision↑ | Recall↑ | Accuracy↑ | MAE↓ | R2↑ | RMSD↓ |
|---|---|---|---|---|---|---|---|---|---|
| GRACE-2L-OAM-L | 26.4M | 0.980 | 6.280 | 0.960 | 1.000 | 0.960 | 0.025 | 0.835 | 0.064 |
| DPA3-3M-FT | 3.27M | 0.987 | 6.369 | **0.987** | 1.000 | **0.987** | 0.019 | 0.901 | 0.069 |
| Nequip-OAM-L | 9.6M | 0.985 | 6.344 | 0.985 | 1.000 | 0.985 | 0.021 | 0.854 | 0.065 |
| Allegro-OAM-L | 9.7M | 0.987 | 6.371 | 0.974 | 1.000 | 0.974 | 0.018 | **0.908** | 0.065 |
| AlphaNet-v1-OMA | 4.65M | 0.965 | 6.312 | 0.980 | 1.000 | 0.980 | 0.020 | 0.882 | 0.079 |
| SevenNet-MF-ompa | 25.7M | 0.970 | 6.346 | 0.985 | 1.000 | 0.985 | 0.019 | 0.888 | 0.064 |
| ORB v3 | 25.5M | 0.964 | 6.307 | 0.964 | 1.000 | 0.964 | 0.021 | 0.860 | 0.075 |
| eqV2 M | 86.6M | **0.988** | **6.382** | 0.976 | 1.000 | 0.976 | **0.015** | 0.904 | 0.069 |
| eSEN-30M-OAM | 30.2M | 0.971 | 6.350 | 0.971 | 1.000 | 0.971 | 0.016 | 0.901 | 0.061 |
| MatRIS-10M-OAM | 10.4M | 0.986 | 6.366 | 0.973 | 1.000 | 0.973 | **0.015** | 0.904 | **0.060** |

We first performed the force field prediction task, in which the model is evaluated on datasets from the molecules, inorganic materials, and catalysis domains, predicting energies, forces, and virials. For fair comparison, we selected MLIPs trained on OMat+sAlex+MPTrj as reference. The results are reported in Table 8. Overall, MatRIS-10M-OAM achieves the best accuracy, and for molecules and inorganic materials, it generally outperforms other models in force prediction.

Table 8: Summary of model performance on LAMBench (Force Field Prediction). Energy (E) RMSE is in meV/atom, Force (F) RMSE in meV/Å, Virial (V) RMSE in meV/atom.

| | GRACE-2L-OAM | | | SevenNet-MF-ompa | | | ORBv3-mpa | | | MatRIS-10M-OAM | | |
|---|---|---|---|---|---|---|---|---|---|---|---|---|
| | E | F | V | E | F | V | E | F | V | E | F | V |
| **Molecules** | | | | | | | | | | | | |
| AIMD-Chig (Wang et al., 2023) | 3.1 | 239.8 | - | 3.4 | 247.9 | - | **2.4** | 200.2 | - | 2.6 | **200.0** | - |
| ANI-1x (Smith et al., 2018) | 32.1 | 365.5 | - | 26.1 | 337.7 | - | **20.0** | 247.9 | - | 22.1 | **246.1** | - |
| MD22 (Chmiela et al., 2023) | 3.5 | 235.9 | - | 4.7 | 238.1 | - | 2.8 | 163.9 | - | **2.4** | **161.5** | - |
| **Catalysis** | | | | | | | | | | | | |
| Vandermause et al. (2022) | **5.5** | 99.4 | 60.6 | 12.7 | 100.8 | **39.1** | 6.8 | **92.6** | 51.2 | 13.8 | 110.9 | 47.8 |
| Zhang et al. (2019) | **251.6** | **723.0** | - | 392.2 | 937.5 | - | 464.9 | 1169.4 | - | 609.2 | 963.4 | - |
| Fernández-Villanueva et al. (2024) | 3.3 | 131.8 | - | 3.0 | 95.6 | - | 2.0 | 86.4 | - | **1.8** | **79.0** | - |
| **Inorganic materials** | | | | | | | | | | | | |
| Lopanitsyna et al. (2023) | **55.4** | 294.7 | 200.1 | **55.4** | 268.8 | 226.9 | 56.4 | 324.2 | **198.2** | 61.0 | **257.2** | 212.9 |
| Lopanitsyna et al. (2023) | 70.1 | 309.7 | 243.1 | 69.8 | 265.3 | 235.8 | 71.0 | 330.6 | 242.4 | 75.6 | 261.2 | 251.6 |
| Batzner et al. (2022) | 0.8 | 111.5 | - | 0.7 | 107.9 | - | **0.6** | 103.7 | - | **0.6** | **103.4** | - |
| Torres et al. (2019) | 3.2 | 161.2 | - | 2.8 | 167.9 | - | 2.8 | **141.9** | - | **2.6** | 148.5 | - |
| Gao et al. (2025) | 17.8 | 135.7 | 164.1 | 17.7 | 104.8 | 161.1 | 20.4 | 167.4 | **141.4** | 14.9 | **83.9** | 150.1 |
| Sours & Kulkarni (2023) | 6.5 | **162.8** | - | 6.5 | 173.6 | - | 6.5 | 182.7 | - | 6.5 | 176.1 | - |

In addition, we also evaluated the performance on property calculation tasks, which include reaction barrier prediction (OC20-NEB task), elastic constant prediction (Elastic Properties task), and molecular conformer energy prediction (Wiggle150 task). The results are reported in Table 9. In the OC20-NEB task, MatRIS-10M-OAM achieves competitive results, but we observe that its pre-

dictions for reaction barriers (Ea) and reaction energies (d) are worse than those of other models, while its success rate remains relatively high. We speculate that this is because MatRIS uses MAE to compute energy loss during training, without placing extra weight on outlier data points, which amplifies their effect in this task; even so, the success rate is not significantly affected. In the elastic and Wiggle150 tasks, MatRIS-10M-OAM achieves the best performance.

Table 9: Summary of model performance on LAMBench (Property Calculation). Metrics are lower-is-better unless otherwise noted.

| Model | OC20-NEB (Wander et al., 2025) | | | | | Elastic Properties | | Wiggle150 (Brew et al., 2025) | |
|---|---|---|---|---|---|---|---|---|---|
| | MAE.Ea (eV) | MAE.d (eV) | Transfer (%) ↓ | Desorption (%) ↓ | Dissociation (%) ↓ | Shear Modulus MAE (GPa) | Bulk Modulus MAE (GPa) | MAE (kcal/mol) | RMSE (kcal/mol) |
| GRACE-2L-OAM | **1.6** | **0.7** | 65.1 | 90.5 | 72.2 | 9.1 | 7.5 | 12.1 | 14.0 |
| SevenNet-MF-ompa | 2.1 | 1.3 | 66.9 | 92.9 | **68.3** | 9.5 | 7.5 | 11.0 | 12.8 |
| ORBv3-mpa | 2.3 | 1.5 | **61.7** | **87.4** | 72.2 | 9.7 | 7.6 | 11.9 | 12.9 |
| MatRIS-10M-OAM | 4.0 | 3.6 | 64.6 | 90.6 | 69.0 | **8.8** | **6.4** | **9.5** | **10.6** |

Overall, the cross-domain evaluation on LAMBench shows that MatRIS-10M-OAM performs consistently well across both force field prediction and property calculation tasks, achieving either the best or highly competitive results, demonstrating strong generalizability and application potential.

### D.3 ZEOLITE BENCHMARK

The Zeolite Dataset (Yin et al., 2025) comprises 16 zeolite structures relevant to catalysis, adsorption, and separation applications. For each type, atomic trajectories were generated via AIMD simulations at 2000 K using VASP. We adopt the pre-partitioned training, validation, and test sets, containing 48,000, 16,000, and 16,000 structures per zeolite, respectively. The model's prediction targets are the total energies and atomic forces of the systems. The results are shown in Table 10; MatRIS achieves SOTA performance overall.

Table 10: MatRIS performance on Zeolite dataset. Energy(**E**) MAE is in **meV**, force(**F**) MAE is in **meV/Å**.

| Type | Deep Pot E | Deep Pot F | AlphaNet E | AlphaNet F | DPA3-L24 E | DPA3-L24 F | MatRIS-M E | MatRIS-M F |
|---|---|---|---|---|---|---|---|---|
| ABWopt | 90 | 90 | 12 | 19 | 13.7 | 19.1 | **3.4** | **6.8** |
| BCTopt | 110 | 50 | 6.8 | 12 | 6.8 | 12.0 | **3.0** | **4.9** |
| BPHopt | 210 | 60 | 29 | **16** | **28.6** | 16.5 | 39.2 | 8.3 |
| CANopt | 150 | 90 | 18 | 15 | 17.3 | 15.1 | **10.2** | **6.5** |
| EDIopt | 40 | 50 | 8.7 | 13.5 | 10.8 | 15.1 | **3.6** | **6.2** |
| FERopt | 290 | 130 | 43 | 28 | 41.6 | 25.4 | **30.6** | **11.9** |
| GISopt | 60 | 50 | 11 | 11 | 12.1 | 11.8 | **6.2** | **5.8** |
| JBWopt | 150 | 70 | 10 | 13 | 9.7 | 12.4 | **4.0** | **5.9** |
| LOSopt | 110 | 70 | 21 | 12 | 21.0 | 13.1 | **17.9** | **6.5** |
| LTAopt | 150 | 64 | **19** | 12 | 21.3 | 13.5 | 20.5 | **7.0** |
| LTJopt | 70 | 110 | 15 | 11 | **13.4** | 11.2 | 14.4 | **6.7** |
| NATopt | 210 | 110 | 16 | 15 | 17.2 | 15.6 | **8.0** | **7.4** |
| PARopt | 90 | 70 | 18 | 24 | 21.4 | 24.2 | **9.5** | **6.8** |
| PHIopt | 60 | 120 | 20 | 14 | 19.6 | 14.4 | **15.9** | **7.5** |
| SODopt | 200 | 110 | 9.5 | 12 | 11.0 | 13.4 | **4.4** | **5.7** |
| THOopt | 160 | 60 | 17 | 15 | 20.2 | 16.9 | **7.5** | **8.0** |

### D.4 DPA2 TEST SET

We evaluate the performance of the MatRIS model on the DPA2 dataset (Zhang et al., 2024) to assess its ability to handle small-scale datasets. This composite dataset integrates 18 domain-specific subsets (e.g., Alloy, Drug, $H_2O$, OC2M) and is generated using various DFT software packages (e.g.,

Table 11: MatRIS performance on DPA2 test sets from Zhang et al. (2024). Energy (**E**) RMSE is in **meV/atom**, force (**F**) RMSE is in **meV/Å**. 'OOM' indicates an Out-Of-Memory error. 'NC' indicates that the model employs a non-conservative force prediction.

| Dataset | Size | GemNet-OC[NC] | | NequIP | | MACE | | eqV2[NC] | | DPA3-L24 | | MatRIS-M | |
|---|---|---|---|---|---|---|---|---|---|---|---|---|---|
| | | E | F | E | F | E | F | E | F | E | F | E | F |
| Alloy | 71K | 14.3 | 85.1 | 44.0 | 175.6 | 16.2 | 190.2 | 8.5 | **62.7** | 7.1 | 99.2 | **5.8** | 96.2 |
| Cathode-P | 59K | 1.5 | 17.9 | 14.3 | 69.8 | 2.6 | 37.8 | 1.1 | 14.9 | 0.6 | 18.6 | **0.3** | **11.9** |
| Cluster-P | 139K | 47.7 | **69.6** | 75.1 | 216.6 | 41.3 | 189.7 | 34.6 | 104.4 | 29.3 | 118.3 | **12.8** | 113.9 |
| Drug | 1380K | 40.5 | 93.6 | 21.6 | 187.2 | - | - | 29.8 | 807.4 | 5.5 | 54.2 | **1.4** | **39.7** |
| FerroEle-P | 7K | 1.5 | 17.9 | 1.1 | 23.0 | 2.3 | 31.7 | 1.1 | 13.0 | 0.3 | 13.4 | **0.2** | **10.9** |
| OC2M | 2000K | 25.0 | 129.1 | 97.4 | 226.1 | - | - | 6.7 | **45.2** | 9.0 | 128.0 | 5.9 | 77.9 |
| SSE-PBE-P | 15K | 2.7 | 8.2 | 1.6 | 41.1 | 1.8 | 29.9 | OOM | OOM | 0.5 | 19.8 | **0.3** | **11.4** |
| SemiCond | 137K | 8.0 | 94.4 | 20.5 | 180.7 | 12.7 | 182.8 | 3.9 | **40.8** | 3.4 | 108.3 | **2.7** | 95.0 |
| H2O-PD | 46K | OOM | OOM | 0.9 | 27.1 | 79.9 | 29.7 | OOM | OOM | 0.4 | 13.7 | **0.3** | **10.7** |
| Ag∪Au-PBE | 17K | 106.0 | 8.0 | 42.3 | 43.8 | 369.1 | 34.5 | 23.4 | **4.4** | **1.1** | 10.9 | 2.5 | 10.7 |
| Al∪Mg∪Cu | 24K | 5.9 | 9.4 | 38.0 | 48.3 | 7.7 | 42.9 | 1.9 | **5.7** | 2.0 | 13.0 | **1.2** | 15.9 |
| Cu | 15K | 6.1 | 5.8 | 6.2 | 16.7 | 38.8 | 13.6 | 1.7 | **3.8** | 1.7 | 7.2 | **0.5** | 6.1 |
| Sn | 6K | 8.4 | 33.7 | 18.2 | 62.2 | - | - | 5.2 | **19.6** | 2.9 | 49.8 | **2.3** | 45.9 |
| Ti | 10K | 44.5 | 87.9 | 27.6 | 137.4 | 8.3 | 94.2 | 19.1 | **48.6** | 4.1 | 91.7 | **3.3** | 80.2 |
| V | 15K | 17.9 | 79.3 | 8.8 | 91.6 | 14.2 | 140.4 | 5.6 | **47.4** | 2.8 | 71.8 | **1.9** | 62.9 |
| W | 42K | 79.1 | 81.2 | 20.8 | 160.4 | 15.6 | 181.2 | 46.8 | **51.3** | 2.5 | 83.3 | **1.9** | 66.3 |
| C12H26 | 34K | 125.8 | 518.7 | 121.4 | 715.6 | 81.9 | 802.3 | 123.1 | 907.4 | 42.4 | 541.6 | **19.5** | **287.6** |
| HfO2 | 28K | 1.2 | 16.1 | 1.5 | 58.8 | 2.3 | 14.7 | 1.0 | **9.1** | 1.4 | 28.9 | **0.3** | 14.6 |

VASP, Gaussian, ABACUS), with the training data for each subset ranging from 6K to 2,000K. We evaluate MatRIS-M using the same training configurations and maintain consistency with the training and test sets as in Zhang et al. (2024) to ensure comparability. More detailed hyper-parameters are reported in Table 12.

All results are summarized in Table 11, with benchmark data for other models taken from Zhang et al. (2024; 2025). Overall, MatRIS-M achieves better accuracy in both energy and force predictions, demonstrating its robustness across diverse domains. Notably, however, eqV2 and GemNet-OC exhibit lower force errors than MatRIS in certain systems. This advantage likely stems from their non-conservative force prediction, where energy and forces are fitted separately at the expense of physical consistency.

### D.5 MD STABILITY EVALUATION

We assessed the stability of MatRIS by examining energy conservation in MD simulations under the NVE ensemble (microcanonical ensemble), using energy drift and thermostatted stability as evaluation metrics. The test data were taken from LAMBench (Peng et al., 2026), including organic molecules and inorganic material systems, which are out-of-distribution for MatRIS-M trained on MPTrj dataset. For each system, atomic velocities were first randomly initialized at 300 K using the Maxwell-Boltzmann distribution. Subsequently, MD simulations were performed in the NVE ensemble for 80 ps with a 1 fs time step.

Figure 9 shows the results of the MD simulations. MatRIS-M is able to conserve energy over long simulations, with both the total energy and the kinetic temperature fluctuating around their initial values, and no large drifts are observed.

### E TRAINING DETAILS AND HYPER-PARAMETERS

We summarize the hyperparameters of the models across different datasets and versions in Table 12. For the MPTrj dataset, we experimented with models of different sizes by varying the number of layers (S, M, and L correspond to 4, 6, and 10 layers, respectively). The learning rate follows a cosine annealing schedule, decaying to the minimum value at the last epoch. In distributed training,

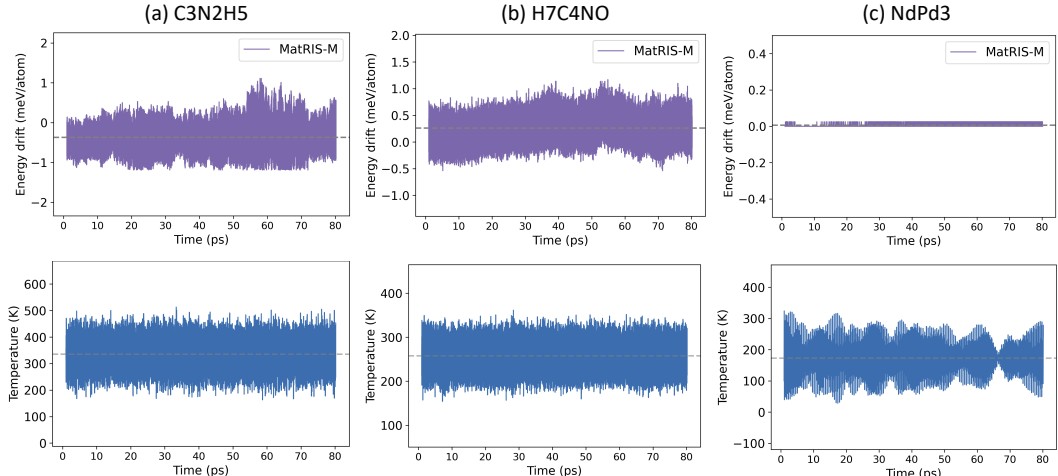

Figure 9: NVE MD simulations of three systems using MatRIS-M. The top row shows the energy drift, and the bottom row shows the time series for the kinetic temperature.

we implemented a load balancing scheme to reduce synchronization overhead (see Section C.1), so the number of samples per GPU may vary; the table reports the average global batch size. Notably, the loss reduction in distributed training is performed using a graph-level scheme to avoid bias toward larger systems (see Section C.2). Additionally, for the Matbench-Discovery benchmark, we pre-trained the MatRIS-M and L models with a denoising process to enhance generalization, with details in Section C.3.

Table 12: Hyper-parameters for MatRIS variants reported in this paper. In MatRIS-OAM, values outside parentheses denote the OMat configuration, those in parentheses the sAlex+MPTrj fine-tuning configuration, and values without parentheses are shared.

| Hyper-parameters | MPTrj-S | MPTrj-M | MPTrj-L | MatRIS-OAM | SPICE-M | MatPES-1.4M | Others |
|---|---|---|---|---|---|---|---|
| Number of MatRIS layers | 4 | 6 | 10 | 10 | 6 | 3 | 6 |
| Dimension of atom features | 128 | 128 | 128 | 128 | 128 | 64 | 128 |
| Dimension of edge features | 128 | 128 | 128 | 128 | 128 | 64 | 128 |
| Dimension of three-body features | 128 | 128 | 128 | 128 | 128 | 64 | 128 |
| Radial basis function | Bessel | Bessel | Bessel | Bessel | Bessel | Bessel | Bessel |
| Number of radial basis functions | 7 | 7 | 7 | 7 | 7 | 7 | 7 |
| Pairwise cutoff | 6.0 | 6.0 | 6.0 | 6.0 | 6.0 | 6.0 | 6.0 |
| Three-body cutoff | 4.0 | 4.5 | 4.5 | 4.5 | 4.0 | 4.0 | 4.0 |
| Global batch size (avg) | 512 | 512 | 320 | 512(320) | 128 | 64 | 1/256 |
| Optimizer | AdamW | AdamW | AdamW | AdamW | AdamW | Adam | AdamW |
| Weight decay | 1e-2 | 1e-2 | 1e-2 | 1e-3 | 1e-3 | 0.0 | 1e-3 |
| Maximum Learning rate | 5e-4 | 3e-4 | 3e-4 | 3e-4(1e-4) | 3e-4 | 3e-4 | 3e-4 |
| Minimum Learning rate | 5e-6 | 3e-6 | 3e-6 | 3e-6(1e-6) | 3e-6 | 3e-6 | 3e-6 |
| Learning rate scheduling | CosLR | CosLR | CosLR | CosLR | CosLR | CosLR | CosLR |
| Number of epochs | 30 | 40 | 100 | 4(8) | 200 | 30 | 100 |
| Loss function | Huber($\delta$=0.01) | Huber($\delta$=0.01) | MAE / L2MAE | MAE / L2MAE | Huber($\delta$=0.01) | MAE | Huber($\delta$=0.01) |
| Gradient clipping norm | 0.5 | 0.5 | 0.5 | 0.5 | 0.5 | 0.5 | 0.5 |
| Energy loss prefactor | 5 | 5 | 5 | 5 | 5 | 5 | 5 |
| Force loss prefactor | 5 | 5 | 5 | 5 | 5 | 5 | 5 |
| Stress loss prefactor | 0.1 | 0.1 | 0.1 | 0.1 | - | 0.1 | 0.1 |
| Magmom loss prefactor | 0.1 | 0.1 | 0.1 | 0.1 | - | 0.1 | - |
| **Denoising Settings** | | | | | | | |
| Corruption probability | - | 100% | 100% | - | - | - | - |
| Maximum number of steps | - | 1000 | 1000 | - | - | - | - |
| Maximum sigma | - | 1.0 | 1.0 | - | - | - | - |
| Minimum sigma | - | 0.0 | 0.0 | - | - | - | - |
| Noise Schedule | - | Linear | Linear | - | - | - | - |
| Number of epochs | - | 20 | 20 | - | - | - | - |

