# OpenReview forum: "MatRIS: Toward Reliable and Efficient Pretrained Machine Learning Interatomic Potentials"
_ICLR.cc/2026/Conference — ICLR 2026 Poster_

### Official Review · Reviewer_npsV · 2025-10-18

**Soundness:** 3
**Presentation:** 1
**Contribution:** 3
**Rating:** 6
**Confidence:** 4

**Summary:**

This work proposes MatRIS, a novel E(3)-invariant model architecture for machine learning interatomic potentials. They adopt dim-wise attention and separable attention for line graphs to encode three-body features. The model outperforms previous SOTAs on Matbench-Discovery, DPA2, and several molecular datasets in the zero-shot setting, performs on par or slightly better than others on MatCalc.

**Strengths:**

1. Using self-attention in the line graph to encode three-body information is novel?, sound, and simple.
2. The separable attention is also with reasonable motivation.
3. The method performs well in several benchmarks.
4. The experiments are sufficient.

**Weaknesses:**

1. Although I assume the methodology is novel, some evidences need to be given with respect to the claim in L94-95: ``$\cdots$, our model is the first to leverage attention mechanisms for modeling three-body interaction''. Specifically, the authors should justify its differences to VisNet [1], FreeCG [2], and MGT [3], which all adopted self-attention mechanisms and captured up to four-body interactions. Is that the case that your model uses self-attention to encode the many-body interactions while they encode many-body interactions without using self-attention in this specific step, but using self-attention to refine the features that have already been encoded? A detailed methodological comparison can be added to the appendix.

2. It appears to be a very rushed paper. Many typographical and presentational errors are spotted in it. I show here some of them:

a) Missing space before almost half of the left parenthesis;

b) Misusing \citep and \cite in many sentences, e.g., ``GemNet Gasteiger et al. (2024) incorporated dihedral angles, and SphereNet Liu et al. (2022) further integrated torsional information. '' in L118-119.

c) Table X or Tab. X. No Table.X, e.g., ``More details on hyper-parameters are given in Table.7 in appendix. Moreover, inspired by (Zaidi et al., 2022; Liao et al., 2024a),'' in L268-269.

d) Omitting math mode, e.g., ``In equivarant models, the predominant strategy for enforcing higher-order equivariance relies on computationally intensive tensor products of rotation order L(Batzner et al., 2022; Batatia et al., 2023; Liao & Smidt, 2023; Liao et al., 2024b).'' in L62-64;

e) The value vector $v$ should be explained in the main text;

f) In Eq. 1, the $j$ is for feature dimension and $i$ for atom index. But in the adjacent context, $i$ and $j$ both represent atom index.

The typographical and presentational errors are too massive to list them all out here. I sincerely recommend the authors go through a complete writing check. I am not that kind of person who rejects a paper solely based on presentations, but the authors could pay more attention to such detail, or it would make a good work fail for such things instead of the methodology itself. It is not worth it for you.

[1] Wang Y, Wang T, Li S, et al. Enhancing geometric representations for molecules with equivariant vector-scalar interactive message passing[J]. Nature Communications, 2024, 15(1): 313.

[2] Shao S, Geng H, Wang Z, et al. FreeCG: Free the Design Space of Clebsch-Gordan Transform for Machine Learning Force Fields[J]. arXiv preprint arXiv:2407.02263, 2024.

[3] Anselmi M, Slabaugh G, Crespo-Otero R, et al. Molecular graph transformer: stepping beyond ALIGNN into long-range interactions[J]. Digital Discovery, 2024, 3(5): 1048-1057.

**Questions:**

1. How can framework be applied to equivariant models? The equivariance, and even the whole invariance will be broken if we manipulate the channel of a geometric tensor individually whose weight higher than 0. Does this framework have the potential to be applied in such scenario?

2. From Eq. 1, the denominator does not contain $x_{ij}$ itself, as it is not in the neighboring list of itself. Is it a typo?

---

> ### Author Response · Authors · 2025-11-21
>
> We thank you for your careful and thorough review, for recognizing the strengths of our method, and for providing many valuable suggestions. We have incorporated all of your comments and ensured that the revised manuscript is free of formatting, typesetting, and spelling errors.
>
> Below, we provide our responses to your questions:
>
> > **Weakness1**: Although I assume the methodology is novel, some evidences need to be given with respect to the claim in L94-95.....
>
> Thank you for bringing up the parts of the manuscript that were unclear. We’ve updated the revised version and added a discussion of the differences between MatRIS and related models—VisNet [1], FreeCG [2], and MTG [3]—in Appendix A.2.
>
>
> > **Weakness2**: It appears to be a very rushed paper. Many typographical and presentational errors are spotted in it...
>
> We carefully checked the manuscript for errors and have corrected all of them, clarified the definitions of related symbols, and ensured that the revised version is free of formatting or typesetting issues.
>
> > **Question1**: How can framework be applied to equivariant models?
>
> We have considered some scenarios for applying our framework to equivariant models. Specifically, we discuss the applications of Dim-wise softmax and Separable attention:
>
> 1. **Dim-wise softmax:** This is applied on invariant features (e.g., features with (L=0)) to compute the attention weights $a_{ij}$, with shape `[edge, 1, D]`.
>
>    The equivariant features $f_{ij}$ are then reshaped to `[edge, (L_max+1)^2, D]`. When performing element-wise multiplication $a_{ij}$ $\times$ $f_{ij}$, the weights $a_{ij}$ are broadcast to match the shape of $f_{ij}$. This multiplication preserves equivariance because channels of different (L) are not mixed, ensuring that equivariance is maintained. This approach is similar to the implementation in eqV1/2 [4,5].
>
>
> 2. **Separable attention:** This can be directly applied to related equivariant attention models (e.g., eqV1/2 [4,5]). Specifically:
>
>    - When generating attention weights, two separate branches compute the source and target attention weights in the same way as in point 1.
>    - The outputs of the two branches are then aggregated separately over the source and target lists to produce the updated node features.
>
> We have added a paragraph in **Section 3.2** discussing the application of our framework to both unconstrained and equivariant models.
>
> > **Question2**: From Eq. 1, the denominator does not contain $x_{ij}$ itself, as it is not in the neighboring list of itself. Is it a typo?
>
> This is not a typo, but a design choice. In the softmax stage, a node gathers information only from its neighbors, excluding itself. Its own information is later added via a learnable residual connection.
>
>
> We’re happy to discuss these points with you, and please feel free to let us know if anything is unclear!
>
>
>
> [1] Wang Y, et al. Enhancing geometric representations for molecules with equivariant vector-scalar interactive message passing[J]. Nature Communications, 2024, 15(1): 313.
>
> [2] Shao S, et al. FreeCG: Free the Design Space of Clebsch-Gordan Transform for Machine Learning Force Fields[J]. arXiv preprint arXiv:2407.02263, 2024.
>
> [3] Anselmi M, et al. Molecular graph transformer: stepping beyond ALIGNN into long-range interactions[J]. Digital Discovery, 2024, 3(5): 1048-1057.
>
> [4] Liao et al. Equiformer: Equivariant Graph Attention Transformer for 3D Atomistic Graphs. ICLR 2023.
>
> [5] Liao et al. EquiformerV2: Improved Equivariant Transformer for Scaling to Higher-Degree Representations. ICLR 2024.

---

> > ### Comment · Reviewer_npsV · 2025-11-21
> > **Thank you**
> >
> > The response has resolved my concerns, and I am happy to raise my score to 8.

---

> > > ### Author Response · Authors · 2025-11-21
> > >
> > > We appreciate your thoughtful review and are pleased that our responses have addressed your concerns. Thank you for your time and valuable feedback.

---

### Official Review · Reviewer_MUqg · 2025-11-01

**Soundness:** 2
**Presentation:** 2
**Contribution:** 3
**Rating:** 4
**Confidence:** 4

**Summary:**

The paper proposes MatRIS, an invariant methodology to construct machine learning interatomic potentials that use three-body attention between edges with a common node.

MatRIS is benchmarked on a broad range of benchmark datasets, including the Matbench Discovery benchmark (for "compliant" tracking fitting only on the MPTrj datasets), MatCalc, and others.

Additionally, the paper reports the computational efficiency of the model.

**Strengths:**

The model, MatRIS, achieves state-of-the-art efficiency of the compliant track of Matbench Discovery with the F1 score of 0.844, and performs equally well on other benchmarks - MatCalc, Molecular Zero Shot Benchmark, and DPA2.

The paper provides several ablations on the model design and fitting procedure.

The computational efficiency of the MatRIS model is competitive in both training and inference.

**Weaknesses:**

The position of the available MLIPs architectures into invariant and equivariant groups is not correct. The paper groups all the models into only two groups, invariant and equivariant, missing the third big group of models - rotationally unconstrained architectures, and assigns many of the rotationally unconstrained architectures into the invariant groups. Because of this, some claims are not entirely true, for instance, "Recent studies (Neumann et al., 2024; Qu & Krishnapriyan, 2024; Rhodes et al., 2025) show that invariant models allow more flexibility in exploring the potential energy surface while being computationally more efficient." The cited papers are unconstrained, not invariant, and thus the claim that ' invariant models allow more flexibility' is hardly correct.

Invariant models are those where all the intermediate representations are invariant with respect to rotations - they are not changed at all if a system is rotated. Example - SchNet.

Equivariant models are those where intermediate representations are invariant or covariant, which means that they transform in a specific manner with respect to the rotation of the input system, such as vectors, Cartesian tensors of higher rank, or spherical tensors.

Unconstrained models are those where there is no restriction on intermediate representations, and they can be learned to transform arbitrarily with respect to rotations.

In general, invariant models are the least expressive and fastest; equivariant models are more expressive and more computationally demanding because of the associated tensor products, and unconstrained models are most expressive and simultaneously fast, roughly as invariant ones, but produce not rotationally invariant predictions, which can sometimes be a problem in molecular dynamics simulations.

The Edge-based design, while claimed to be new, has already been proposed in the community. Examples are Atomistic Line Graph Neural Network, Point Edge Transformer, and many others.

Similarly, there are multiple attention-based architectures with O(N) scaling that apply attention to local neighborhoods.

**Questions:**

Could you clarify the discussion in light of arranging models into invariant, covariant, and unconstrained ones?

Could you provide more insights into the reasons why MatRIS is so computationally efficient?

---

> ### Author Response · Authors · 2025-11-21
>
> We sincerely thank you for your careful review. We have made major revisions to the manuscript based on your suggestions (highlighted in blue) to improve its rigor. Below are our responses to the concerns and issues you raised:
>
> >**Weakness**：About architectural novelty.
>
>
> Thank you for pointing this out. MatRIS has some technical differences compared with related models, as discussed in Appendix A. The main differences are summarized as follows:
>
> 1. **Dim-wise softmax:** The attention weights vary across feature dimensions, allowing the model to differentiate the importance of each dimension.
> 2. **Separable Attention:** This mechanism considers both directions—from source to target and from target to source—and generates separate attention weights for each direction.
> 3. **Explicit and efficient modeling:** With (O(N)) complexity, MatRIS can efficiently and explicitly encode and update three-body features, enhancing the model capacity.
>
> > **Question1 & weakness**: Could you clarify the discussion in light of arranging models into invariant, covariant, and unconstrained ones?
>
> Following your suggestion, we discussed the technical approaches of invariant, equivariant, and unconstrained MLIPs in the “Related Works” section (revisions highlighted in blue). In addition, we corrected the imprecise wording in the revised manuscript.
>
>
>
> > **Question2** : Could you provide more insights into the reasons why MatRIS is so computationally efficient?
>
> The high computational efficiency of MatRIS can be attributed to the following optimizations:
>
> 1. **Load-balancing strategy:** We found that atomic datasets often exhibit a long-tail distribution[1,2,3]. In distributed training, this can lead to uneven workloads. To address this, we designed a load-balancing method using ‘dynamic batching’ to avoid workload imbalance. The detailed procedure is described in Appendix C.1. This approach achieves a speedup of **2.05–2.63×**.
> 2. **Kernel optimization:** We observed that certain operators used in double-backward, especially SiLU and LayerNorm, are computationally inefficient, and few existing works optimize these computations. We applied optimizations to the second-order derivative computations of these operators, further accelerating training.
>
> The speedups achieved by the two optimization methods are as follows:
>
> |                     | Step-by-step speedup | Cumulative speedup |
> | ------------------- | -------------------- | ------------------ |
> | Baseline            | 1.0                  | 1.0                |
> | Load-balancing      | 2.05～2.63           | 2.05～2.63         |
> | Kernel optimization | 1.62～1.66           | **3.32～4.35**     |
>
> By doing these optimizations, MatRIS-S achieves a final speedup of **3.32×**, while MatRIS-L reaches **4.35×**. In addition, we will make these optimized kernels and the load-balancing code an open-source library for the community to use.
>
>
> Thank you again for your review. Please let us know if you have any questions. We would be happy to respond.
>
> [1] Deng et al. Chgnet as a pretrained universal neural network potential for charge-informed atomistic modelling. Nature Machine Intelligence.
>
> [2] Schmidt et al. Improving machine-learning models in materials science through large datasets. Materials Today Physics.
>
> [3] Barroso-Luque et al. Open materials 2024 (omat24) inorganic materials dataset and models. arXiv preprint arXiv:2410.12771 .

---

### Official Review · Reviewer_KaLv · 2025-11-02

**Soundness:** 3
**Presentation:** 3
**Contribution:** 3
**Rating:** 4
**Confidence:** 5

**Summary:**

The paper introduces MatRIS, a new invariant universal machine learning interatomic potential that achieves high accuracy at a lower computational cost than existing equivariant models. Instead of relying on tensor operations to enforce symmetry, MatRIS uses an attention-based architecture that explicitly models three-body interactions through a line graph and a separable linear attention mechanism with O(N) complexity. MatRIS matches or outperforms state-of-the-art models on benchmarks like Matbench-Discovery and MatPES. Ablation studies confirm that components like dimwise softmax, separable attention, and learnable envelopes significantly improve performance.

**Strengths:**

1. The architecture is new. The authors introduce a new architectural approach by explicitly modeling three-body interactions using a line graph attention mechanism. Each edge in the atom graph becomes a node in the line graph, enabling the model to attend over bond angles. This is claimed to be the first use of attention for three-body interactions in ML potentials, extending beyond traditional message-passing that usually considers only pairwise interactions.
2. They employ a separable attention mechanism that splits the message-passing into two parallel branches: one capturing how each neighbor affects a central atom, and another for how the central atom influences its neighbors. Also, this "attention" is implemented with linear complexity in the number of atoms.
3. The proposed model demonstrates competitive or superior results on a few benchmarks.

**Weaknesses:**

1. The evaluation on large-scale datasets is limited. A notable gap in the experiments is the absence of results on the latest massive and diverse datasets, specifically Open Materials 2024 (OMat 24) and Open Molecules 2025 (OMol 25). If the model is claimed to be scalable, then it should perform well on these datasets. At least a benchmark on OMol 4M would be nice to have.
2. Despite the terminology, MatRIS’s mechanism differs from standard scaled dot-product self-attention with Q/K/V projections and global token mixing. Weights are computed per-dimension over local neighbors (a “dim-wise softmax”) and derived by applying a linear map to edge features, and followed by separate target/source softmax normalizations. In other words, the “attention” acts more like edge-conditioned, per-channel neighbor weighting than full QKV attention, and it is local (achieving linear O(N) complexity by avoiding all-pairs interactions). This design brings efficiency but may limit the capacity for global, content-based mixing compared to traditional Transformer attention.
3. MatRIS is built on local, cutoff-based neighborhoods (pairwise and three-body/angle graphs), which means long-range electrostatics and other delocalized effects are not explicitly modeled or evaluated. To demonstrate robustness in regimes where long-range interactions matter (e.g., biomolecules, electrolytes), it would be great to test on OMol 25 and report results on biomolecule / electrolyte validation splits.
4. The paper frames MatRIS as a universal MLIP, but the core pretraining and evaluations are materials-centric: Matbench trained on MPTrj, MatCalc using MatPES/MPTrj, and results on DPA2 materials test sets. While the abstract highlights “universal” scope and mentions a “Molecular dataset”, the molecular side is limited to small zero-shot style checks rather than large-scale molecular pretraining/evaluation. And there's no mention on other types of chemical systems, such as surface, organic crystals, polymers, MOFs, etc.

**Questions:**

1. What unique signals does the source/target split capture that a single asymmetric aggregator cannot? A controlled ablation that matches parameters and FLOPs would be nice (or does your ablation matched parameters and FLOPs?)
2. In low-data regimes, how does MatRIS compare to strong equivariant baselines?
3. Since forces/stress are energy-conservative, how does MatRIS behave in long NVE MD runs (energy drift, thermostatted stability)?

---

> ### Author Response · Authors · 2025-11-21
> **Part1: Response weaknesses**
>
> Thank you for your review. We are glad that you recognize the strengths of our model, and we will incorporate your suggestions into the manuscript. Below are our responses to the concerns and questions you raised:
>
> > **Weakness 1**: The evaluation on large-scale datasets is limited.
>
> We have added results on a large-scale dataset (OMat24) and compared MatRIS-10M-OAM with mainstream models on existing benchmarks. Overall, MatRIS-10M-OAM achieves competitive or superior performance.
> The newly added experiments are as follows:
>
> **1.Matbench-Discovery benchmark （non-compliant）**
>
> |      Model      | Param. |      F1      |     DAF      |     MAE      |      R2      |     RMSD     |
> | :------------- | :----: | :----------: | :----------: | :----------: | :----------: | :----------: |
> | **Full splits** |        |              |              |              |              |              |
> |     eqV2 M      | 86.6M  |    0.896     |    5.243     |    0.020     |    0.842     |    0.069     |
> |  eSEN-30M-OAM   | 30.2M  | 0.902 |  **5.281**   |  **0.018**   | 0.860 | 0.061 |
> | MatRIS-10M-OAM  | 10.4M  |  **0.903**   | 5.275 | 0.019 |  **0.864**   |  **0.060**   |
> |   **Unique prototypes**    |        |              |              |              |              |              |
> |     eqV2 M      | 86.6M  |    0.917     | 6.047 |    0.020    |    0.848     |    0.069     |
> |  eSEN-30M-OAM   | 30.2M  |  **0.925**   |  **6.069**   |  **0.018**   | 0.866  | 0.061 |
> | MatRIS-10M-OAM  | 10.4M  | 0.921 |    6.039     | 0.019 |  **0.871**   |  **0.060**   |
> We have included the full results in Appendix D.1.
>
> **2.MatCalc Benchmark**
>
> |     Model      |      d       |      E       |      K      |      G      |     CV      |   f/f_DFT    |
> | :------------ | :----------: | :----------: | :---------: | :---------: | :---------: | :----------: |
> |     eqV2 M     |  **0.235**   |  **0.017**   |    25.4     |    17.5     |    80.4     |  **0.999**   |
> |  eSEN-30M-OAM  | 0.299 |    0.089     | 11.9 | 14.8 | 4.35 | 0.996 |
> | MatRIS-10M-OAM |    0.316     | 0.025 |  **10.6**   |  **13.3**   |  **3.97**   |    0.985     |
> We have added the full results to Table 2 in Section 4.2.
>
> **3.MDR benchmark**
>
> |     model      | $\omega_{max}$ |    S     |    F     |    CV    |
> | :------------ | :---------- | :------ | :------ | :------ |
> | SevenNet-ompa  |  15   |  **8**   | 4 | 3 |
> |  eSEN-30M-OAM  |  15   |    10    | 4 | 3 |
> | MatRIS-10M-OAM |    **8**     | 9 |  **3**   |  **2**   |
> We have added the full results to Figure 4 in Section 4.3.
>
> 4. **LAMBench benchmark**
>
>    The full results have been added in Appendix D.2.
>
>
> > **Weakness 2&3**: Concerns about model capacity and the ability to capture global information
>
> Thank you for your insight. The design of MatRIS is intentional:
>
>
> - Regarding concerns about **“whether it limits model capacity”**:
>
>  Investigating the expressive power of $O(N)$ attention compared to dot-product attention is a topic worth further study. To our knowledge, there is currently no definitive conclusion. We refer to the ablation studies in **Equiformer [1]**, which found that using **$O(N)$ attention** provides **better computational efficiency and more expressive** than dot-product attention. Based on this idea and considerations of efficiency, we adopted this $O(N)$ attention mechanism.
>
> - Regarding concerns about “**long-range interaction capability**”:
>
> Like most MLIPs (e.g., eSEN [2], eqV1/V2 [1,3], DPA series [4], etc.), MatRIS first constructs a graph based on a truncated neighborhood and then propagates global information through the message-passing mechanism. For example, in MatRIS-L, the cutoff radius is 6.0 Å and there are a total of 10 stacked GNN layers, which allows it to capture information up to 60 Å.
>
> Additionally, MatRIS supports existing plugins such as Latent Ewald Summation (LES) [5], which can explicitly introduce long-range interactions. However, to ensure a fair comparison with other models, LES was not used in our experiments.
>
>
>
> > **Weakness 4**: The paper frames MatRIS as a universal MLIP, but the core pretraining and evaluations are materials-centric....
>
> We evaluated MatRIS-10M-OAM on LAMBench, which includes more diverse test datasets and more tasks. Overall, MatRIS-10M-OAM performs well across these tasks. The full results have been added in Appendix D.2.

---

> > ### Author Response · Authors · 2025-11-21
> > **Part2: Response questions**
> >
> > > **Question1**:What unique signals does the source/target split capture that a single asymmetric aggregator cannot? A controlled ablation that matches parameters and FLOPs would be nice...
> >
> > The source/target split allows a node to capture its dual role as both a “source” and a “target” in a directed graph, which cannot be achieved by a single asymmetric aggregator. A single aggregator typically lets a node only capture information from incoming edges as a “target,” whereas the split architecture enables aggregation in both roles: as a “target,” the node aggregates information from incoming edges, and as a “source,” it aggregates information from outgoing edges.
> >
> > For example, consider a neighbor pair 0–1, forming two directed edges $e_{01}$ and $e_{10}$ with weights $a_{01}$ and $a_{10}$, respectively. When updating node 0, a single aggregator can only use the weight $a_{10}$ to aggregate information from the incoming edge $e_{10}$, ignoring the outgoing edge $e_{01}$. In contrast, the source/target split uses both $a_{10}$ and $a_{01}$, allowing node 0 to aggregate incoming information as a “target” and outgoing information as a “source,” thereby enhancing model capacity.
> >
> > In our ablation studies, the number of parameters was controlled to be around 4M.
> >
> >
> >
> > > **Question 2**: In low-data regimes, how does MatRIS compare to strong equivariant baselines?
> >
> > We followed the evaluation protocol used in [6], using the Liquid Water and three ices dataset [7]. Only 133 samples (<0.1%) were used for training, while the test set contains about 140,000 samples. Compared with the NequIP [6] results, the outcomes are as follows:
> >
> > | System       | NequIP (Energy RMSE) | NequIP (Force RMSE) | MatRIS  (Energy RMSE) | MatRIS  (Force RMSE) |
> > | ------------ | -------------------- | ------------------- | --------------------- | -------------------- |
> > | Liquid Water | 1.6                  | 51.4                | **0.9**                   | **48.5**                 |
> > | Ice Ih(b)    | 2.5                  | 57.8                | **1.1**                   | 57.7                 |
> > | Ice Ih(c)    | 3.9                  | **29.1**                | **1.2**                   | 77.2                 |
> > | Ice Ih(d)    | 2.6                  | 24.1                | **1.2**                   | **15.4**                 |
> >
> > Overall, MatRIS demonstrates competitive performance in low-data regimes.
> >
> >
> >
> > > **Question 3**: How does MatRIS behave in long NVE MD runs?
> >
> > We used MatRIS-M (MPTrj-trained) for NVE simulations, selecting samples provided by LAMBench [8], including organic molecules and inorganic material systems (which are out-of-distribution for the MPTrj-trained model). We found that the total energy and kinetic temperature showed no drift.
> >
> > These results have been added to Appendix D.5 of the manuscript.
> >
> > Please let us know if you have any feedback to our explanation or other questions. We will be very happy to respond.
> >
> > [1] Liao et al. Equiformer: Equivariant Graph Attention Transformer for 3D Atomistic Graphs. ICLR 2023.
> >
> > [2] Fu et al. Learning Smooth and Expressive Interatomic Potentials for Physical Property Prediction. ICML 2025.
> >
> > [3] Liao et al. EquiformerV2: Improved Equivariant Transformer for Scaling to Higher-Degree Representations. ICLR 2024.
> >
> > [4] Zhang et al. A Graph Neural Network for the Era of Large Atomistic Models.
> >
> > [5] King et al. Machine learning of charges and long-range interactions from energies and forces. Nature Communications 2025.
> >
> > [6] Batzner et al. E(3)-equivariant graph neural networks for data-efficient and accurate interatomic potentials. Nature Communications 2022.
> >
> > [7] Zhang et al. Deep potential molecular dynamics: a scalable model with the accuracy of quantum mechanics. Phys. Rev. Lett.
> >
> > [8] Peng et al. LAMBench: A Benchmark for Large Atomistic Models. arXiv preprint arXiv:2504.19578 .

---

> ### Comment · Reviewer_KaLv · 2025-11-28
> **Response to Authors**
>
> Thanks the authors for the detailed reply. I’m convinced by the OMat and MD results. I will raise my score to 6. (I don’t have access to edit my score right now, will do it after I regain access)
>
> I still have two further concerns that prevents me to increase my score further:
> 1. The long range argument is not convincing to me. GNN is known to have over smoothing or over squashing problem, so one cannot just calculate the radius in that way. This is also the reason why the eSEN model performs worse than GemNet in the OMol dataset. I was hoping to see some improvements in LR, if the model could utilize its efficiency and increase the cutoff by quite a lot, and lead to good performance on OMol. Even a proof of concept run on OMol 4M would be great.
> 2. The model size is relatively small (10M). I would be curious to see how would the model scale with more parameters. Predictive scaling would be essential for the actual usage of the model.

---

> > ### Author Response · Authors · 2025-12-03
> >
> > We thank the reviewer for acknowledging our additional experiments and for the attention to long-range effects. Due to limited rebuttal time and computational resources, we focused on training on the large-scale OMat24 dataset and conducting experiments most relevant to the paper’s core conclusions.
> >
> >
> >
> > Our main focus is on improving the efficiency and performance of pretrained MLIP in modeling local interactions, as supported by our existing evaluations. While long-range interactions are an interesting direction, they are beyond the scope of this work, as noted in the “Conclusion and Future Work” section.
> >
> >
> >
> > We have also conducted preliminary scalability tests (parameters 4M → 10M, data 1.58M → 100.8M), which show clear scaling trends. Further scaling experiments are beyond the rebuttal timeframe, but the current results on OMat24 support that our architecture is scalable.

---

### Author Response · Authors · 2025-11-21
**Update Manuscript**

We sincerely thank all reviewers for their valuable reviews! We have updated the manuscript accordingly. All revisions are highlighted in blue in the revised version. A summary of the changes is as follows:

1. According to Reviewer **npsV**’s suggestion, we analysis the technical differences between our model and prior works (in Appendix A), and discuss the possbility on our proposed method applying to unconstrained and equivariant models (in Section 3.2). The notation of Eq.1, the typos and typographical errors have been carefully checked and revised.
2. According to Reviewer **MUqg**’s suggestion, we revised the Related Works (Section 2), corrected some typos and imprecise expressions, and discussed the differences between MatRIS and related works in Appendix A.
3. According to Reviewer **KaLv**’s suggestion, we evaluate MatRIS on OMat24 dataset, benchmark on MatCalc, MDR benchmark, Matbench-Discovey, LAMBench ( in Sections 4.2/4.3， Appendix D.1, D.2), and conduct NVE MD downstream tasks (in Appendix D.5).

---

### Author Response · Authors · 2025-12-03
**Summary of Rebuttal**

Dear Program Chairs, Senior Area Chairs, and Area Chairs,

In this work, we introduce MatRIS, an invariant MLIP that leverages separable attention to model three-body interactions. Across multiple benchmarks, MatRIS achieves SOTA or near-SOTA performance while maintaining computational efficiency.



We thank the reviewers for their time and constructive feedback. We have substantially revised the manuscript to address all concerns. Below, we summarize the main updates and clarifications:



**Strengths acknowledged by the reviewers:**

- **Novelty of the architecture**: Reviewers npsV and KaLV recognized the innovation of our architectural design and the soundness of its motivation.
- **Competitive performance**: All reviewers acknowledged that our method achieves strong accuracy across multiple benchmarks while maintaining computational efficiency.
- **Sufficiency of experiments**: Reviewer npsV considered our experiments sufficient. In addition, following Reviewer KaLV’s suggestion, we further included evaluations on large-scale datasets.



**Reviewers’ concerns:**

- **Presentation of the paper (Reviewers npsV, MUqg)**: We carefully checked the paper for spelling, formatting errors, and some imprecise expressions, and corrected them to ensure that the revised paper is free of such issues.
- **Comparison with existing work (Reviewers npsV, MUqg)**: The reviewers suggested providing evidence to support the novelty of our method. In the revised version, we added a comparison in Appendix A that highlights relevant works. We also clarify the differences from existing methods.
- **Evaluation on large-scale datasets and additional downstream tasks (Reviewer KaLV)**: We trained our model on the large-scale OMat24 dataset and evaluated it across multiple benchmarks. We also evaluated the model on additional downstream tasks (LAMBench, MD), showing that our method  performs well in these benchmarks.



All reviewers are highly knowledgeable in the field of MLIPs. After our responses and revisions, Reviewer npsV indicated that they would raise their rating to 8. Reviewer MUqg has not yet responded, but we believe our replies addressed their concerns. Reviewer KaLV plans to raise their rating to 6, and the factor limiting a higher score is the modeling of long-range interactions, which is interesting but beyond the scope of this work.



Overall, we believe this is a meaningful contribution. We also thank all reviewers for their comments; we have actively addressed all concerns and revised the paper, further improving its quality.

We sincerely thank the Area Chair and the reviewers for their time and thoughtful reviews.

Sincerely,
The Authors

---

### Meta-Review · Area_Chair_dC18 · 2026-01-02

**Summary:**

The paper proposes MatRIS, a novel and efficient invariant machine learning interatomic potential.

Reviewer MUqg was primarily concerned with the paper's novelty and presentation. While I agree that edge-based attention itself is not a new concept, its specific application to machine learning potentials appears to be novel. This sentiment was shared by Reviewer npsV, who also agreed that the methodology is innovative.

MatRIS achieves a Pareto-optimal tradeoff between efficiency and accuracy, matching or exceeding state-of-the-art methods while maintaining high computational efficiency. Two reviewers specifically highlighted the model's strong performance. Furthermore, during the rebuttal phase, the authors provided convincing results on a larger dataset, further validating the model's scalability.

In summary, the method is novel and demonstrates Pareto-optimal performance against strong baselines. It directly addresses the critical challenge of the high training costs associated with large-scale interatomic potential datasets. I am pleased to recommend the paper for acceptance.

**Reviewer Concerns:**

Key concerns:

- Reviewers npsV and KaLV asked for further discussion of the novelty of the paper. The Authors rebuttal has successfuly addressed this concern.
- Reviewer MUqg was concerned with novelty of the paper: edge-based attention has been proposed in prior works. Though according to Authors’ claims not in the context of machine learning potentials.
- Reviewer KaLV asked for experiments on larger datasets. The Reviewer was convinced with the results on a new dataset and raised the score.  Some concerns regarding long range interactions were still raised but the method does seem to achieve competitive performance on a wide range of benchmarks.

**Reviewer Scores:**

Two reviewers have raised their scores (to 6 and 8). Reviewer MUqg gave score of 4, mostly concerned with novelty. It is unclear if after Authors’ rebuttal the Reviewer would have increased the score.

---

### Decision · Program_Chairs · 2026-01-26

Accept (Poster)